# Enhancing MIMO Spatial-Multiplexing and Parallel-Decoding under Interference by Computational Feedback

**Avner Elgam** [1,*] , **Yossi Peretz** [2] **and Yosef Pinhasi** [1]

1 Faculty of Engineering, Ariel University, Ariel 40700, Israel
2 Department of Computer Sciences, Lev Academic Center, Jerusalem College of Technology, Jerusalem 9372115, Israel
* Correspondence: avner.elgam@msmail.ariel.ac.il

**Abstract:** In this paper, we propose a new digital Hard-Successive-Interference-Cancellation (HSIC), the Alternating Projections-HSIC (AP-HSIC), an innovative fast computational feedback algorithm that deals with various destructive phenomena from different types of interferences. The correctness and convergence of the proposed algorithm are provided, and its complexity is given. The proposed algorithm possesses the functionality of canceling digital interference without the aid of physical feedback between the receiver and the transmitter or the loading of learning information about the state of the Multiple Input–Multiple Output (MIMO) channel to the transmitter. The proposed AP-HSIC algorithm enables a parallel decoding process from the parallel transmission of Orthogonal-Space–Time-Block-Coding (OSTBC) under a complex and challenging wireless environment to facilitate the Dynamic Spectrum Sharing (DSS) capability. We present a performance comparison of the proposed algorithm with the algorithm for Multi-Group-Space–Time-Coding (MGSTC) under MIMO fading channels and general interference or high-level Additive White Gaussian Noise (AWGN). Mathematical analysis and real-time simulations show the advantages of the proposed algorithm compared to the MGSTC decoding algorithm.

**Keywords:** MIMO; MGSTC; interference; Alternating Projections; Hard-Successive-Interference-Cancellation

## 1. Introduction

Dynamic and shared-spectrum-access techniques are among the most challenging approaches in advanced modern wireless communication. Examples of technologies implementing dynamic spectrum spatial capabilities are 5th-Generation New-Radio (5G-NR)-Heterogeneous-Network (HetNet) [1,2], 6th-Mobile-Generation [3], and Wi-Fi-6,7,8 802.11.ax-be. Moreover, dealing with multi-interference signals, Over-the-Air (OTA) in the domains of space, time, and frequency, or coping with jamming attacks that strongly correlate with transmission signals, and simultaneously integrating the application of modern MIMO techniques poses challenging issues. Existing solutions to these problems include: intelligent access [4,5], intelligent SIC [6–11], smart management sharing systems, and dynamic-cognitive-radio-spectra [12–14].

More and more systems in both the civilian and military fields are required to deal with multi-path phenomena and diverse sources of interference in conjunction with ultra-reliability and low-air transport requirements [2]. In the advanced modern communication field, several users per urban area and the number of mobile operators conducting Stand-Alone (SA) unlicensed communications are massive and intensifying. This process of congestion leads to the destruction of wireless channels, causing bad conditions for wave propagation [15], scattering phenomena [16], and interference-jamming problems [17–19]. The meaning of the destruction of wireless channels is that there is a strong correlation between the desired signal and the interference signal that causes a significant reduction in the complex power gain of the wireless links. In designing next-generation communication

systems, it is necessary to consider these decoding and fairness problems in addition to dealing with these destructive effects.

Two main approaches have been developed in recent years to deal with the challenges of offsetting the destructive effects of various sources of interference. The capabilities of those approaches combine spatial selectivity and wireless communication techniques that increase the channel's capacity and bring about accurate and fast decoding processes. The two main approaches can be classified as follows: classical conventional and non-conventional analog–digital-SIC approach [7,20] or None-Orthogonal-Multiple-Access (NOMA) techniques [21–23], and advanced MIMO-beamforming techniques such as analog–digital hybrid-beamforming structure [24–26], and multi-layer-precoding full-dimensional massive MIMO systems [14,27].

The first approach includes three different families of decoders incorporating interference offset algorithms. The first family has algorithms such as classical analog–digital-SIC [7,20,28], zero forcing-interference cancellation (ZF-IC) [29], MGSTC [29–31], and SIC based on machine-learning/deep-learning algorithms such as SICnet [28] and-deepSIC [7]. The main advantage of these techniques is the absence of the requirement for physical feedback between the receiver and the transmitter to share the information evaluation of the wireless channel with the transmitter (e.g., the Channel-State-Information-at-the-Transmitter (CSIT)). The common denominator of this family is based on serial process decoders designed for the spatial separation of a block of information intended for a specific user from a system of simultaneously transmitted blocks. The coalition of the remaining blocks considered internal interferences are serially offset to decode the desired block. These methods are limited in dealing with interference processes from external interferers, such as neighboring users who produce a strong correlation with the signals of the desired transmitter or an intelligent jammer activated in the same geographic-spatial space. The reason for this weakness is that the external disturbance process causes a violation of the legality of the internal offset process predefined in the SIC algorithm. This process damages the iterative decoding process and significantly damages the system's performance (as we prove in the next section).

The second family is the NOMA technique. The NOMA is based on multi-carrier power control, which allows identifying each user's reception power values or power coefficients according to their differences in power-multiplexing. This method is also vulnerable to external interference from outside the network or from the source of an intelligent jammer. These interferences disrupt the spatial separation process, which is based on differentiating the reception power of each user.

The third family is based on minimum-mean-square detection interference-rejection combiner (MMSE-IRC) [32]. The conventional MMSE-IRC enables spatial selectivity, as well as dealing with neighboring users who produce complex disturbances within the legal framework of the network, a high order of diversity, and advanced spatial plural-ism. The MMSE-IRC is limited in the aspect of eliminating the outsourcing of dynamic interference and dealing with destructive wireless effects because the construction of the system mechanisms is heavily based on None-Stand-Alone (NSA) networks or on network-based regulation protocols, with the help of Physical-Upload-Share-Channel (PUSHC), and auxiliary Demodulation-Reference-Signals (DM-RS) [1,2].

The second approach is high-resolution beamforming techniques [33], hybrid beam-forming [4,24], analog–digital-beamforming, and multi-layer-precoding full-dimensional massive MIMO systems [34]. This approach's disadvantage is that the receiver must share with the transmitter CSIT via physical feedback that might be vulnerable. As a result, the ability of an intelligent jammer or a general disturbance to damage the CSIT is natural and exists in physical reality. The balance between the need to develop systems without closed feedback and the development of advanced feedback-based methods is derived from considerations of the need to implement real-time communication, communication with minimal computational overhead, and constraints of physical conditions that cannot connect physical feedback between the receiver and the transmitter. Examples of such

systems with physical constraints and limitations are satellite communication systems, critical wireless links (for example, medical systems based on wireless communication), and autonomous vehicles.

In this paper, we propose a new digital interference cancellation algorithm—the AP-HSIC—that only requires the Channel-State-Information-at-the-Receiver (CSIR) assumption and generates computational feedback at the receiver. This computational feedback can overcome the effects of random scatters in a multi-path fading channel and quasi-static Flat-Rayleigh-Fading MIMO channels, combined with high-level AWGN and general interference scenarios. We compare the MGSTC decoding algorithm's performance based on the MIMO array's serial-decoding mode and the proposed algorithm, AP-HSIC, which combines parallel processing decoding methods in the MIMO array. This comparison was conducted in an environment of different interferers. The comparison is reflected in the performance levels of Bit–Error Ratio (BER) vs. the Signal-to-Noise Ratio (SNR) or BER vs. Signal-to-Interference Ratio (SIR).

The analysis considers different constellation orders (Quadrature Phase Shift Keying (QPSK); 8-PSK; and 16-PSK) and challenges the two systems under scenarios of high-level AWGN and two kinds of interference: Partial Band Noise (PBN) [35] and general interference [1]. The presence of general interfering signals can be associated with neighboring users or jamming attacks [17].

The proposed AP-HSIC can generate significant capabilities for successfully canceling digital interference. The AP-HSIC offers flexibility regarding stand-alone (SA) networks without the request to control a sharing channel that statistically measures the spatial domain. The receiver also does not require any control or physical feedback from the transmitter. In addition, AP-HSIC can decode symbols in a parallel MIMO mode in real-time without slowing down the decoding process and simultaneously with the interference cancellation process. These features are significant in applying ultra-bit-rate and heavy-capacity channel requirements. It has the ability to discern between the original channel response matrix and interfering factors. The next section defines these factors as the interrupting part, $\Delta H$, where we show how to calculate an approximation to $\Delta H$ and perform an online update to the general MIMO channel response matrix. The computation of $\Delta H$ allows a total offset of the interference signal. The result is that the system can decode the originally transmitted symbols without increasing the power and re-transmitting. We may assume statistical dependence between the general interference (or the jamming signal) and the user signal, as well as statistical dependence between the AWGN and the user signal. Furthermore, with the knowledge of the CSIR, we can distinguish between a signal that transmits data from an interference signal without changing to pilot symbols, changing modulation, or increasing the transmission power.

The common denominator of advanced wireless communication systems that include interference offset techniques and innovative parallel decoding algorithms is that the network operates in Time-Division-Duplex (TDD) communication mode. A significant advantage of this mode is in the abstraction of the ability to evaluate the wireless channel and reduce the information overhead between the transmitter and the receiver. The significant and growing problem with this mode is that it is weak against interference or neighboring users in the space. The considerable advantages of the proposed algorithm are not only that it allows information to be decoded in a parallel manner at the same time as the offset of the interferers without the need for physical feedback between the transmitter and the receiver, but it also allows applicability to communication systems that are unable to produce physical feedback due to considerations of system architecture, critical response time, etc. Examples of this applicability are satellite systems, autonomous vehicles, and ultra-low latency wireless systems. The proposed algorithm allows these systems to transmit high data rates in a high order of modulations and simultaneously decode them under diverse interfering scenarios and under TDD mode.

The remainder of the paper is organized as follows: Section 2 presents the proposed communication MIMO model under various interference scenarios. Section 3 presents the

AP-HSIC algorithm, where we prove its correctness, prove its convergence, and analyze its complexity. Section 4 describes the real-time simulations and numerical results based on SIMULINK and MATLAB platforms and provides a comparison of the performances of both systems: the MGSTC and the AP-HSIC. In Section 5, we derive conclusions and a vision for further research on efficient solutions to communication system problems under interfering scenarios.

## 2. Proposed Communication MIMO Model under Various Interference

This section describes and analyzes a proposed communication MIMO model under various interference scenarios for two different architectures. The first architecture includes a transmitter based on the MGSTC scheme, including the diversity-transmitting technique-OSTBC. The receiver is based on the MGSTC decoding algorithm. This architecture is described in [29]. The MGSTC has the property of interference cancellation and decoding capability with the ability to separate a single symbol information block from a set of transmitted blocks and combines an error-reducing mechanism. The MGSTC technique works by iterating a serial decoding mode in the receiver. It includes the parallel transmission of a series of blocks in space and, on the receiver side, a serial decoding capability that separates the various transmitted blocks.

In this architecture, as we mentioned, the MGSTC decoding algorithm decodes and de-multiplexes the individual data streams in a specific user from the other streams and creates, through serial iterations (i.e., multi-stage decoding), a communication mode capable of offsetting the noise part (a sum of AWGN with the rest of the broadcast blocks). This process is performed by utilizing the accuracy of a single decoding stream relative to the last decoding iteration.

There are two distinct advantages of the MGSTC decoding algorithm. The first one is based on the fact that, at the end of each iteration, when we move to the next group order, the MGSTC scheme produces a diversity gain for every matrix of OSTBC transmission modulation symbols, $S_{c_i}$ ($S_{c_i}$ included $c_i$ MGSTC component-block as $i = 1, \ldots, I$). The diversity gain is increased by $n_i \times (n_1 + \cdots + n_i + N_r - N_t)$ when the $i$ is the iteration number and $n_i$ is the number of transmission antennas in the $i$'th group, while $N_r$, and $N_t$ are the number of total receiver antennas and the number of total transmission antennas, respectively.

Having the ability to raise the diversity order means being able to reduce the average power transmission of each group inversely to the diversity order of each group; for example, the average power per symbol transmission at antennas 1 and 2 in the first group is defined as $E_s$, and the diversity order is 4, the average power per symbol at antennas 3 and 4 will be $\frac{E_s}{2}$, with diversity order of 8. The second advantage is based on the fact that in each given iteration, as the iteration process continues forward, the decoding accuracy increases, a process that results in the elimination of the noise part in relation to the previous iteration.

The challenging issue in the MGSTC and in multi-stage decoding, in general, is that the advantages become disadvantages in several communication scenarios. Those advantages, as mentioned above, at the same time, lead to disadvantages for MGSTC systems when interference cases are present in the same spatial domain. The disadvantages, reflected in the SIC process of the MGSTC, are based on it being a serial decoding mode of data blocks. Every error accumulated in a given SIC process is dragged into the next iteration and amplified. Another significant weakness of the MGSTC is the wireless links between the antenna couples in the descending order of the antenna array of the transmitter (e.g., antennas 5 and 6 in our simulation) to all antenna arrays of the receiver. The weakness arises from the fact that these paths are more attenuated in terms of the SNR or SIR and because of the reliance on diversity gain. Thus, in the presence of interfering signals, fading effects, or high levels of AWGN that hit these paths, an imbalance in the trade-off between a higher diversity gain and lower SNR occurs, leading to increased values of BER.

We propose the second architecture using the MGSTC-OSTBC at the transmitter and the AP-HSIC scheme at the receiver instead of the MGSTC decoding algorithm. This

proposed hybrid scheme produces immunity against various interference scenarios, as can be seen in the simulation results below. In addition, a more efficient and accurate process of parallel spatial decoding of the series of simultaneously transmitted symbols blocks is achieved in this scheme compared to the first scheme we described.

We separately analyze and simulate the two architectures under three challenging interference scenarios acting in the same spatial domain. The first interference scenario is a high-leveled AWGN. The second scenario is PBN, and the third scenario is a general interference simulating a smart jamming device or a neighboring user.

We start the first and essential analysis with the general interference case. In that case, the communication standard ideal model $Y = HS_{c_i} + Z$ changes to (see [29]):

$$Y = \sqrt{P}H_{TR}S_{c_i} + \sqrt{\frac{P}{SIR}}H_{JR}S_J + \sqrt{\frac{P}{SNR}}Z, \tag{1}$$

where $Y$ is the received signal, with dimensions $[N_r \times k]$, $N_r$ is the number of receiver antennas, and $k$ is the number of sample symbols per frame. The MIMO channel response matrix between the transmitter and receiver is $H_{TR}$, with dimensions $[N_r \times N_t]$, where $N_t$ is the number of transmission antennas. The MIMO channel-response matrix between the interference and the receiver is defined by $H_{JR}$, with dimensions $[N_r \times N_J]$, where $N_J$ is the number of interference transmission antennas. $S_{c_i}$ is the matrix of OSTBC transmission modulation symbols of the desired transmission, with dimensions $[N_t \times k]$ and $S_J$ is the matrix of OSTBC interference modulation symbols, with dimensions $[N_J \times k]$. Finally, the $SIR$ is the signal interference ratio, $P$ is the total transmission power, and $Z$ is the independent complex Gaussian random variable noise. The model communication is described in Figure 1.

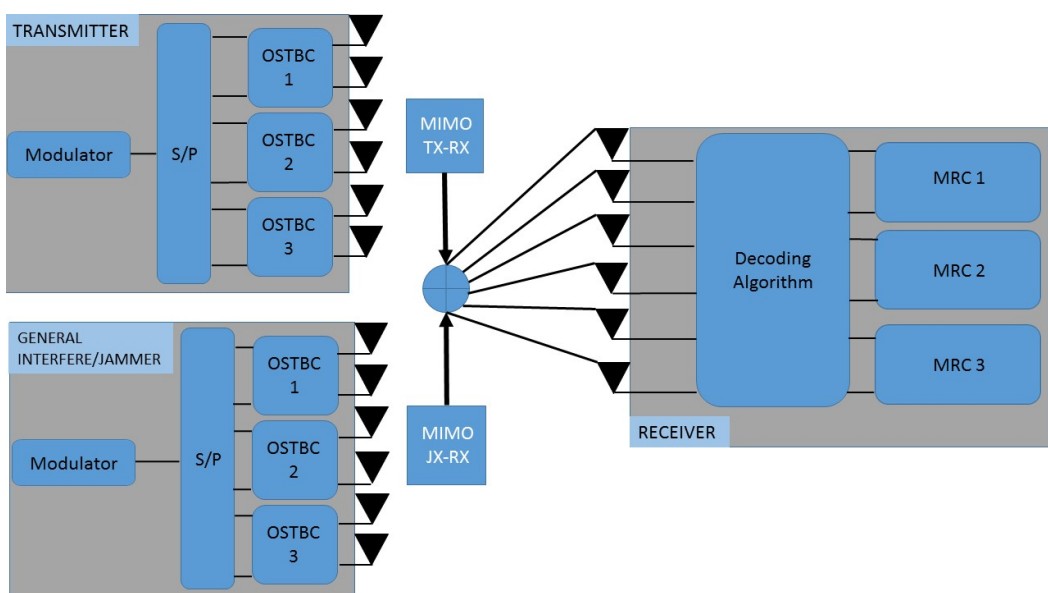

**Figure 1.** Scheme blocks of the MGSTC transmitter, general receiver (including MGSTC or AP-HSIC decoding algorithm), and general interference/smart jammer.

For the MGSTC decoding algorithm, in the presence of the interference signal, $\widetilde{Y}_{c_1}$, [29] becomes:

$$\begin{aligned}
\widetilde{Y}_{c_1} = \Theta_{c-c_1}Y &= \Theta_{c-c_1}\left[\sqrt{P}H_{TR}S_{c_i} + \sqrt{\frac{P}{SIR}}H_{JR}S_J + \sqrt{\frac{P}{SNR}}Z\right] \\
&= \sqrt{P}\Theta_{c-c_1}H_{TR}S_{c_i} + \Theta_{c-c_1}\sqrt{\frac{P}{SIR}}H_{JR}S_J + \sqrt{\frac{P}{SNR}}\widetilde{Z}_{c-c_1}
\end{aligned} \tag{2}$$

where $\widetilde{Y}_{c_1}$ is the receive-matrix for decoding the first symbol-block, $S_{c_1}$. $\Theta_{c-c_1}$ is the null space matrix relative to the decoding process of $S_{c_1}$, and $\widetilde{Z}_{c-c_1}$ is the independent complex Gaussian random variable noise multiplied with the null space matrix, $\Theta_{c-c_1}$.

The second part of this equation, the interference that is received as an additive factor to the MGSTC decoding algorithm, produces the most destructive effects as it destroys the orthogonality of $\Theta_{c-c_1}$ relative to the other part of the MIMO channel-response matrix–destruction that the MGSTC algorithm cannot deal with (see [36]).

In order to solve the problem of interference estimation and digital interference cancellation in the presence of general interference by using self-computational feedback at the receiver, we describe two main approaches with different assumptions to model the interference intervention. The first approach is based on the assumption that the number of interference transmission antennas and the number of legitimate transmission antennas are different, implying that there exists no correlating matrix $\Psi$ such that $S_J = \Psi S_{c_i}$, and therefore we do not have any information about the interference tactic. The second approach is applied under the assumption that there exists a correlation unknown matrix $\Psi$ such that $S_J = \Psi S_{c_i}$. Under the most general assumption, writing $S_{c_i} = S$, we can decompose (1) into:

$$
\begin{aligned}
Y &= \sqrt{P} H_{TR} S + \sqrt{\frac{P}{SIR}} H_{JR} S_J + \sqrt{\frac{P}{SNR}} Z \\
&= \sqrt{P} H_{TR} S + \sqrt{\frac{P}{SIR}} H_{JR} S_J (I - S^+ S + S^+ S) + \sqrt{\frac{P}{SNR}} Z \\
&= \left( \sqrt{P} H_{TR} + \sqrt{\frac{P}{SIR}} H_{JR} S_J S^+ \right) S + \sqrt{\frac{P}{SIR}} H_{JR} S_J (I - S^+ S) + \sqrt{\frac{P}{SNR}} Z,
\end{aligned}
\tag{3}
$$

where $S^+$ is the Moore–Penrose pseudo-inverse of $S$. Multiplying from the right by $S^+ S$, we obtain:

$$
Y S^+ S = \left( \sqrt{P} H_{TR} + \sqrt{\frac{P}{SIR}} H_{JR} S_J S^+ \right) S + \sqrt{\frac{P}{SNR}} Z S^+ S,
$$

since $S^+ S S^+ = S^+$ (which implies $(I - S^+ S) S^+ S = 0$, see Remark 1) and since $S S^+ S = S$.

Let the interference factor be defined by:

$$
\Delta H = \sqrt{\frac{P}{SIR}} H_{JR} S_J S^+.
\tag{4}
$$

Furthermore, let

$$
H = \sqrt{P} H_{TR},
\tag{5}
$$

and $\tilde{Z} = \sqrt{\frac{P}{SNR}} Z S^+ S$.

Now, $\widehat{Z} := \sqrt{\frac{P}{SNR}} Z \sim \mathcal{CMN}_{m \times k} \left( 0_{m \times k}, \frac{\sigma}{m} I_m, \frac{\sigma}{k} I_k \right)$, where $\sigma = \sigma_{\widehat{Z}} = \sqrt{\frac{P}{SNR}}$, which implies that $\tilde{Z} \sim \mathcal{CMN}_{m \times k} \left( 0_{m \times k}, \frac{\sigma}{m} I_m, \frac{\sigma}{k} S^+ S \right)$. We therefore have

$$
\sigma_{\widetilde{Z}}^2 = \frac{\sigma^2}{k} trace\left( S^+ S \right) \leq \sigma^2,
\tag{6}
$$

since $trace(S^+ S) = rank(S^+ S) = rank(S) \leq N_t$ and $N_t \leq k$ (whatever the coding modulation is, using the fact that $S^+ S$ is Hermitian with simple eigenvalues from the set $\{0, 1\}$). The last implies that we reduce the noise part by multiplying with $S^+ S$, assuming that $k$ is large enough.

We therefore have

$$
Y S^+ S = (H + \Delta H) S + \tilde{Z}.
\tag{7}
$$

Note that when a correlation matrix $\Psi$ exists such that $S_J = \Psi S$, then $S_J(I - S^+S) = \Psi S(I - S^+S) = 0$ and (3) is equivalent to

$$Y = \left( \sqrt{P}H_{TR} + \sqrt{\frac{P}{SIR}}H_{JR}S_J S^+ \right) S + \sqrt{\frac{P}{SNR}}Z, \qquad (8)$$

which we write as:

$$Y = (H + \Delta H)S + \tilde{Z}, \qquad (9)$$

where $\tilde{Z} = \sqrt{\frac{P}{SNR}}Z$.

In order to compute $\Delta H$ and $S$ from (7) or from (9), we can start with the standard ideal model to obtain a first approximation for $S$, using the known channel response matrix $H$. Then, we iterate on (7) or on (9) to compute the best approximations for $\Delta H$ and for $S$, as explained in Section 3. In [36], it has been proven and demonstrated through simulations that the worst-case scenario is where a rotation correlation matrix $\Psi$ exists such that $S_J = \Psi S$. Thus, our analysis includes the case where $S, S_J$ are correlated. Note that $\|S^+S\|_F^2 = trace\big((S^+S)^*S^+S\big) = trace(S^+S) = rank(S) \leq N_t$. Therefore:

$$\begin{aligned}
\|YS^+S - (H + \Delta H)S\|_F &= \|YS^+S - (H + \Delta H)SS^+S\|_F \\
&= \|(Y - (H + \Delta H)S)S^+S\|_F \leq \|Y - (H + \Delta H)S\|_F \cdot \|S^+S\|_F \\
&\leq \|Y - (H + \Delta H)S\|_F \cdot \sqrt{N_t}.
\end{aligned}$$

This implies that $\|YS^+S - (H + \Delta H)S\|_F \leq \epsilon$, if $\|Y - (H + \Delta H)S\|_F \leq \frac{\epsilon}{\sqrt{N_t}}$. Thus, continuing with (9) does not lose the generality, as (7) can be solved by the same algorithm, with the error threshold $\frac{\epsilon}{\sqrt{N_t}}$ instead of $\epsilon$.

A more simplified case is when $S_J = S$ (although it is not simple for the jammer to generate $S$ exactly). Practical examples of intelligent jammers (or neighboring users with the same modulation order), where $S_J = S$ or $S_J = \Psi S$, were considered in [17,36,37].

Note that, in view of (3), the part $\sqrt{\frac{P}{SIR}}H_{JR}S_J(I - S^+S)$, which is orthogonal to $S^+S$ (in the Frobenius inner-product $\langle A, B \rangle_F = tr(B^*A)$) is simply canceled by multiplication with $S^+S$ from the right. However, the part $\Delta HS = \sqrt{\frac{P}{SIR}}H_{JR}S_J S^+S$, which is in the direction of $S^+S$, cannot be canceled without intervening in the channel-state by using some feedback. This explains why, in the aforementioned papers, the simplified cases appear as the most challenging interferences. Mathematically, these resemble simple cases or mathematical simplification, but physically, these cases create disturbances in terms of nonlinear distortions in the constellation or duplication of symbols in the constellation, as described in [36].

In the following, we present computational feedback that cancels $\Delta H$ by computing $\Delta H$ and using it to reliably decode $S$ and update the new MIMO channel-response matrix to be $H + \Delta H$ without the need to update the transmitter. The next interference scenario is the PBN case. The PBN case represents the noise energy caused by a jammer, a neighboring user under the same operator, or a user from a neighboring operator that leaks energy across a specific portion of the system's target bandwidth. The noise power bandwidth of the PBN may be less than that of the user, but the power amplitude of the PBN can be much higher than the user's received signal. The last scenario is a high-leveled AWGN that is added to the existing noise part, $Z$, as described in [38].

## 3. The Alternating Projections HSIC Algorithm

In this section, we write $n = N_t$ for the number of transmission antennas and $m = N_r$ for the number of receiver antennas for the simplicity of presentation. Let $A$ be any $\mathbb{C}^{m \times n}$ matrix. Then, we denote the Moore–Penrose pseudoinverse by $A^+$. Let an singular value decomposition (S.V.D.) of $A$ be denoted by $A = U\Sigma V^*$, where $U$ is an $m \times m$

unitary matrix, $V$ is an $n \times n$ unitary matrix, and $\Sigma$ is an $m \times n$ rectangular diagonal matrix with non-negative entries, i.e., $\sigma_1 \geq \sigma_2 \geq \ldots \geq \sigma_\ell > 0$, where $\ell \leq \min(m,n)$ are the singular values of $A$. Then, $A^+ = V\Sigma^+ U^*$, where $\Sigma^+$ is just the diagonal $n \times m$ matrix with $\frac{1}{\sigma_1}, \ldots, \frac{1}{\sigma_\ell}$ on its main diagonal. Note that, for $A = U\Sigma V^*$, as above, we have $A = \sum_{k=1}^\ell \sigma_k u_k v_k^*$, where $u_k, v_k, k = 1, \ldots, \ell$ are the first $\ell$ columns of $U, V$, respectively. Therefore, $A^+ = \sum_{k=1}^\ell \frac{1}{\sigma_k} v_k u_k^*$.

**Remark 1.** *Note that $A^+ = \lim_{\epsilon \to 0^+} A^*(\epsilon I + AA^*)^{-1}$. Therefore, if $A$ is a random matrix, and $\widetilde{A} = A$ Almost Surely (A.S.), then $\widetilde{A}^+$ always exists and, $\widetilde{A}^+ = A^+$ A.S. In that case, we have: $\mathbb{E}\left[\widetilde{A}\right] = \mathbb{E}[A]$ and $\mathbb{E}\left[\widetilde{A}\right]^+ = \mathbb{E}[A]^+$, as well as $\mathbb{E}\left[\widetilde{A}^+\right] = \mathbb{E}[A^+]$ and $\mathbb{E}\left[\widetilde{A}^+\right]^+ = \mathbb{E}[A^+]^+$. Let $L_A = I_n - A^+A$ and $R_A = I_m - AA^+$. Then, $AL_A = 0$, $L_A A^+ = 0$ and $A^+ R_A = 0$, $R_A A = 0$, where $L_A^2 = L_A$, $R_A^2 = R_A$ and $L_A, R_A$ are self-adjoined with simple eigenvalues in the set $\{0,1\}$. Furthermore, $(AA^+)^2 = AA^+$ and $(A^+A)^2 = A^+A$ are self-adjoined with simple eigenvalues in the set $\{0,1\}$.*

Let $z$ be a random $m \times 1$ complex vector. Then, we write $z \sim \mathcal{CN}_{m \times 1}(\mu, \Sigma)$ if the p.d.f. of $z$ is $\frac{\exp\left(-\left((z-\mu)^* \Sigma^{-1}(z-\mu)\right)\right)}{\det(\pi\Sigma)}$, where $\mu = \mathbb{E}[z]$ is an $m \times 1$ complex vector, and $\Sigma = \mathbb{E}\left[(z - \mu)(z - \mu)^*\right]$ is a positive-definite $m \times m$ complex matrix. Let $Z$ be a random $m \times n$ complex matrix. Let $vec(Z)$ denote the function transforming the matrix $Z$ into a vector by aligning its columns into a single $mn \times 1$ column vector. We write $Z \sim \mathcal{CMN}_{m \times n}(M, U, V)$ for $Z$ having the Probability Distribution Function (P.D.F.):

$$\frac{\exp\left(-trace\left(V^{-1}(Z-M)^* U^{-1}(Z-M)\right)\right)}{\pi^{mn} \det(U)^n \det(V)^m} = \frac{\exp\left(-\left\|U^{-1/2}(Z-M)V^{-1/2}\right\|_F^2\right)}{\det(\pi V \otimes U)},$$

where $M = \mathbb{E}[Z]$ is a complex $m \times n$ matrix, $V$ is a positive-definite complex $n \times n$ matrix, and $U$ is a positive-definite complex $m \times m$ matrix. One can show that $Z \sim \mathcal{CMN}_{m \times n}(M, U, V)$ if and only if $vec(Z) \sim \mathcal{CN}_{mn \times 1}(vec(M), V \otimes U)$. Note that, if $Z \sim \mathcal{CMN}_{m \times n}(M, U, V)$, then $N + AZB \sim \mathcal{CMN}_{m \times n}(N + AMB, AUA^*, B^*VB)$. Also note that if $Z \sim \mathcal{CMN}_{m \times n}(M, U, V)$, then $\mathbb{E}\left[(Z - M)(Z - M)^*\right] = U trace(V)$, and $\mathbb{E}\left[(Z - M)^*(Z - M)\right] = V trace(U)$. The latter implies that

$$\sigma^2 := \mathbb{E}\left[\|Z - M\|_F^2\right] = \mathbb{E}\left[trace(Z - M)^*(Z - M)\right]$$
$$= trace\mathbb{E}\left[(Z - M)^*(Z - M)\right] = trace(U)trace(V).$$

Therefore, if $Z \sim \mathcal{CMN}_{m \times n}\left(0_{m \times n}, \frac{\sigma}{m}I_m, \frac{\sigma}{n}I_n\right)$, then, $\mathbb{E}\left[\|Z\|_F^2\right] = \sigma^2$. The following lemma can be easily proven.

**Lemma 1.** *Let $A \in \mathbb{C}^{m \times n}$ and $B \in \mathbb{C}^{m \times p}$. The matrix equation $AX = B$ has solutions if and only if $AA^+B = B$ (i.e., $R_A B = 0$). In that case, the set of all solutions is given by:*

$$X = A^+B + L_A W, \tag{10}$$

*where $W \in \mathbb{C}^{n \times p}$ is an arbitrary $n \times p$ matrix. Moreover, we have $\|X\|_F^2 = \|A^+B\|_F^2 + \|L_A W\|_F^2$, implying that a minimal Frobenius-norm solution is $X = A^+B$. Moreover, even when the condition $AA^+B = B$ is not satisfied, $X = A^+B$ is still a minimal Frobenius-norm solution to the problem $\min_{X \in \mathbb{C}^{n \times p}} \|AX - B\|_F^2$.*

The following theorem would be useful for turning our non-convex problem into a convex one:

**Theorem 1.** *The set of full-rank matrices is dense in the set of all matrices, i.e., for any $A \in \mathbb{C}^{m \times n}$ and any $\epsilon > 0$, there exists a full-rank matrix $A_\epsilon$ such that $\|A - A_\epsilon\|_F \leq \epsilon$.*

The proof of this theorem and of the following theorems is given in the Appendix A.
Under the assumptions mentioned above, the standard ideal MIMO model is described by:

$$Y = HS + Z, \tag{11}$$

where $S$ is an $n \times k$ OSTBC symbol matrix of signals. We assume that $k \geq m \geq n$. The $m \times n$ matrix $H$ is the channel matrix (random with the unknown p.d.f., but assumed to be stationary) and $Z \sim \mathcal{CMN}_{m \times k}\left(0_{m \times k}, \frac{\sigma}{m} I_m, \frac{\sigma}{k} I_k\right)$, which is equivalent to $vec(Z) \sim \mathcal{CN}_{mk \times 1}\left(0_{mk \times 1}, \frac{\sigma^2}{mk} I_k \otimes I_m\right)$.

**Theorem 2.** *Assume that the communication is made with some modulation not containing 0 and let $r_0$ be the radius of the modulation point with minimal radius. Assume that $k \geq \max(m, n)$ and let $S = \begin{bmatrix} S_1 & S_2 & \cdots & S_\ell & S_{\ell+1} \end{bmatrix}$ be the OSTBC symbol matrix of signals, where $k = n\ell + q$ and each $S_j$ is $n \times n$, chosen such that $S_j S_j^* \geq r_0^2 I_n$ for $j = 1, \ldots, \ell$, and $S_{\ell+1}$ is an arbitrary $n \times q$ matrix. In addition, we also assume that $S$ is known to the receiver. Let $H_0 = YS^+$. Then, $H_0$ is a random matrix such that:*

$$\mathbb{E}\left[\|H - H_0\|_F^2\right] = \frac{\sigma^2}{k}\|S^+\|_F^2. \tag{12}$$

*Moreover, since $\|S^+\|_F$ is bounded above, by taking $k \to +\infty$, it follows that $\mathbb{E}\left[\|H - H_0\|_F^2\right] = 0$, implying that $H_0 = H$ A.S. holds.*

In view of Remark 1, $H_0 = H$ A.S. implies that $H_0^+ = H^+$ A.S., from which we can conclude that $\mathbb{E}[H_0] = \mathbb{E}[H]$, and that $\mathbb{E}[H_0^+] = \mathbb{E}[H^+]$. Now, when $S$ is unknown, when we search for $S$ that minimizes $\|Y - HS\|_F^2$, by Lemma 1, we need to take $S = H^+Y$. However, as $H$ is random and $Y$ is random, $S = H^+Y$ is random, while the true $S$ that was sent by the transmitter is not random. However, as it is almost sure that $H_0 = H$, we can take $\widetilde{S} = \mathbb{E}[H_0^+]Y$ as an approximation for $S$. Note that $\widetilde{S}$ is still a random matrix, but it would give us what is needed, as is expressed in the following theorem.

**Theorem 3.** *Let $H_0$ be an $m \times n$ matrix as in Theorem A2, and further assume that $m \geq n$ and that $\mathbb{E}[H_0]$ is full-rank. Let $S$ denote an unknown signal matrix that is sent by the transmitter and let $Y = HS + Z$ be the signal measured by the receiver. Let $\widetilde{S} = \mathbb{E}[H_0]^+Y = \mathbb{E}[H_0]^+(HS + Z)$ (note that $\widetilde{S}$ is a random variable matrix, while $S$ is not random). Then,*

$$\mathbb{E}\left[\widetilde{S}\right] = S. \tag{13}$$

The following theorem connects the expectation of the square error between the approximated computed symbol matrix $\widetilde{S}$ and the real symbol matrix $S$, in terms of the transmission power, the number of antennas, and the SNR level. This would give us a new formula for the BER, as we will see in the simulations section.

**Theorem 4.** *Let $Y = HS + Z$, where $H$ can be the ideal channel matrix or $H + \Delta H$, the resulting channel matrix of the disrupted channel. Assume that the exact $H$ (or the exact $H + \Delta H$, respectively) is known to the receiver and assume that $H$ (or $H + \Delta H$, respectively) is constant for the current session. Let $S$ be the true symbol matrix for the current session, and let $\widetilde{S} = H^+Y$ be the approximated symbol matrix. Then, the distribution of $\widetilde{S} - S$ (given the constant matrix $H$ or $H + \Delta H$, respectively) is:*

$$\widetilde{S} - S \sim \mathcal{CMN}_{n \times k}\left(0_{n \times k}, \frac{\sigma}{m}H^+H^{+*}, \frac{\sigma}{k}I\right). \tag{14}$$

*Moreover, the expectation of the square error satisfies:*

$$\mathbb{E}\left[\left\|\widetilde{S} - S\right\|_F^2\right] = \frac{\|H\|_F^2 \|H^+\|_F^2}{mSNR} = \frac{P\|H^+\|_F^2}{mSNR}. \tag{15}$$

In the following, we propose an algorithm for the computational self-feedback of the receiver that identifies both $\Delta H$ and an unknown matrix symbol $S$. For this purpose, we regard $H_0$ as $H$ and ignore the measurement noise $Z$. Therefore, we write $Y = HS$. Assume that the system has undergone a jamming attack, and the new channel matrix is $H + \Delta H$. Now, the signal received by the receiver is:

$$Y = (H + \Delta H)S. \tag{16}$$

In view of Theorem A1, we may assume w.l.o.g. that $H + \Delta H$ is full-rank, i.e., has $rank(H + \Delta H) = n$, since we assume that $n \leq m$. Therefore, $(H + \Delta H)^+(H + \Delta H) = I_n$, and (16) implies that

$$(H + \Delta H)^+ Y - S = 0. \tag{17}$$

Let us write (17) as:

$$\begin{bmatrix} S & (H + \Delta H)^+ \end{bmatrix} \begin{bmatrix} I_k \\ -Y \end{bmatrix} = 0. \tag{18}$$

Then, in view of Lemma 1, the general solution to the last equation is given by:

$$\begin{bmatrix} S & (H + \Delta H)^+ \end{bmatrix} = \begin{bmatrix} W_1 & W_2 \end{bmatrix} \left( I_{m+k} - \begin{bmatrix} I_k \\ -Y \end{bmatrix} \begin{bmatrix} I_k \\ -Y \end{bmatrix}^+ \right), \tag{19}$$

where $W_1, W_2$ are arbitrary matrices with sizes $n \times k$ and $n \times m$, respectively. Since

$$\begin{bmatrix} I_k \\ -Y \end{bmatrix}^+ = (I_k + Y^*Y)^{-1} \begin{bmatrix} I_k & -Y^* \end{bmatrix},$$

it follows that:

$$\begin{cases} S = W_1 \left( I_k - (I_k + Y^*Y)^{-1} \right) + W_2 Y (I_k + Y^*Y)^{-1} \\ (H + \Delta H)^+ = W_1 (I_k + Y^*Y)^{-1} Y^* + W_2 \left( I_m - Y(I_k + Y^*Y)^{-1} Y^* \right), \end{cases} \tag{20}$$

which is equivalent to:

$$\begin{cases} S = W_1 Y^* (I_m + YY^*)^{-1} Y + W_2 (I_m + YY^*)^{-1} Y \\ \quad = (W_1 Y^* + W_2)(I_m + YY^*)^{-1} Y \\ (H + \Delta H)^+ = W_1 Y^* (I_m + YY^*)^{-1} + W_2 (I_m + YY^*)^{-1} \\ \quad = (W_1 Y^* + W_2)(I_m + YY^*)^{-1}, \end{cases} \tag{21}$$

which is more efficient from the complexity point of view since we assume that $m \leq k$. Note that the set (21) is actually a subspace, that is the left kernel of $\begin{bmatrix} I_k & -Y^T \end{bmatrix}^T$ that we denote as $Ker\left( \begin{bmatrix} I_k & -Y^T \end{bmatrix}^T \right)$ and is, therefore, closed and convex. Note that $W_1 = 0$ and $W_2 = H^+(I_m + YY^*)$ leads to $\Delta H = 0$ and $S = H^+Y$, that is the best approximation for the true $S$ in terms of Lemma 1 and Theorem A3 when there is no interference.

Now, as (21) might yield an infeasible point related to the symbol coding that is being used (here, we use QPSK, 8-PSK, and 16-PSK), letting $QPSK = \{\pm 1 \pm i\} \subset \mathbb{C}$, we define a projection $P_{QPSK} : \mathbb{C}^{n \times k} \to QPSK^{n \times k}$ by (22) below with $\ell = 2$, which actually projects any entry of the matrix $S$ onto the closest point in the $QPSK$ constellation (breaking ties arbitrarily). In the following experiments, we used Algorithm 1, also with 8-PSK and 16-PSK

codings, for performance comparison with the QPSK coding. For this purpose, we defined the following general projections for $2^\ell$-PSK (for any $\ell \geq 2$, including $2^2$-PSK=QPSK):

$$\begin{cases} P_{2^\ell - PSK}(S)_{p,q} = \cos(\phi) + i\sin(\phi) \\ \phi = \dfrac{2\pi\kappa}{2^\ell}, \ \kappa = \left\lfloor \dfrac{2^\ell \theta}{2\pi} + \dfrac{1}{2} \right\rfloor \\ \theta = \arctan\left(\Im(S_{p,q}), \Re(S_{p,q})\right) \end{cases} \tag{22}$$

In order to use the following Algorithm 1, we need to encapsulate $QPSK^{n \times k}$ in a relatively small closed convex set, for which projections can be conveniently calculated. Since any point in $QPSK$ has an absolute value $\sqrt{2}$, we define by $\mathbb{B}\left(0, \sqrt{2nk}\right)$ the matrix-ball of all $n \times k$ matrices $S$ with $\|S\|_F \leq \sqrt{2nk}$. Now, if $S \in QPSK^{n \times k}$, then $|S_{i,j}| = \sqrt{2}$ implying that $\|S\|_F^2 = \sum_{i=1}^n \sum_{j=1}^k |S_{i,j}|^2 = 2nk$, from which we conclude that $S \in \mathbb{B}\left(0, \sqrt{2nk}\right)$ and moreover, that each such matrix is on the boundary of the matrix-ball.

---

**Algorithm 1: AP-HSIC: Receiver Self-Feedback Algorithm**

---

**Require:** *An algorithm for computing Moore–Penrose Pseudoinverses and algorithms for computing $P_{\mathcal{C}_0}$, $P_{\mathcal{C}_1}$ and $P_{QPSK}$.*

**Input:** $H, Y, \epsilon > 0$.

**Output:** $\Delta H$ and $S$ such that $(H + \Delta H)^+ Y = S$, where $S \in QPSK^{n \times k}$.

1. $\Delta H_0 \leftarrow 0$
2. $S_0 \leftarrow H^+ Y$
3. $\left((H + \Delta H_1)^+, S_1\right) \leftarrow P_{\mathcal{C}_1}\left((H + \Delta H_0)^+, S_0\right)$
4. $\left((H + \Delta H_1)^+, S_1\right) \leftarrow P_{\mathcal{C}_0}\left((H + \Delta H_1)^+, S_1\right)$
5. $\Delta H_1 \leftarrow \left((H + \Delta H_1)^+\right)^+ - H$
6. $t \leftarrow 1$
7. **while** $\sqrt{\|\Delta H_t - \Delta H_{t-1}\|_F^2 + \|S_t - S_{t-1}\|_F^2} > \epsilon$ **do**
8.     $\left((H + \Delta H_{t+1})^+, S_{t+1}\right) \leftarrow P_{\mathcal{C}_1}\left((H + \Delta H_t)^+, S_t\right)$
9.     $\left((H + \Delta H_{t+1})^+, S_{t+1}\right) \leftarrow P_{\mathcal{C}_0}\left((H + \Delta H_{t+1})^+, S_{t+1}\right)$
10.     $\Delta H_{t+1} \leftarrow \left((H + \Delta H_{t+1})^+\right)^+ - H$
11.     $t \leftarrow t + 1$
12. **end while**
13. $\Delta H \leftarrow \Delta H_t$
14. $S \leftarrow P_{QPSK}(S_t)$
15. **return** $t, \Delta H, S$

---

Let

$$\mathcal{C}_0 = \left\{ \left((H + \Delta H)^+, S\right) \mid \Delta H \text{ arbitrary and } S \in \mathbb{B}\left(0, \sqrt{2nk}\right) \right\}, \tag{23}$$

and

$$\mathcal{C}_1 = \left\{ \left((H + \Delta H)^+, S\right) \mid \Delta H \text{ and } S \text{ are given by (21)} \right\}, \tag{24}$$

and note that these sets are closed and convex sets in $\mathbb{C}^{n \times m} \times \mathbb{C}^{n \times k}$.

We now need projections on $\mathcal{C}_0$ and on $\mathcal{C}_1$ such that any given point in $\mathbb{C}^{n \times m} \times \mathbb{C}^{n \times k}$ which is out of the set is projected to the closest point in the boundary of the set. Let $P_{\mathcal{C}_0}$ be defined by $(A, B) \mapsto (A, \beta B)$, where:

$$\beta = \begin{cases} 1 & \text{if } \|B\|_F \leq \sqrt{2nk} \\ \dfrac{\sqrt{2nk}}{\|B\|_F} & \text{otherwise} \end{cases}$$

The projection $P_{C_0}$ is realized in Algorithm 2.

For a given couple $(A, B)$ as above, let $f(W_1, W_2) = \left\| (W_1 Y^* + W_2)(I_m + YY^*)^{-1} - A \right\|_F^2$ and let $g(W_1, W_2) = \left\| (W_1 Y^* + W_2)(I_m + YY^*)^{-1} Y - B \right\|_F^2$.

Let $U = (U_1, U_2) \in \mathbb{C}^{n \times m} \times \mathbb{C}^{n \times k}$ be a directional matrix, which is with $\|U\|_F = \sqrt{\|U_1\|_F^2 + \|U_2\|_F^2} = 1$ and let $h > 0$. Then, the directional derivative at $(W_1, W_2)$ in the direction $U$ is defined by:

$$\nabla_U f(W_1, W_2) = \frac{\partial f}{\partial U}(W_1, W_2) = \lim_{h \to 0^+} \frac{f(W_1 + hU_1, W_2 + hU_2) - f(W_1, W_2)}{h}. \tag{25}$$

We compute:

$$
\begin{aligned}
f(W_1 + hU_1, W_2 + hU_2) - f(W_1, W_2) &= \left\| ((W_1 + hU_1)Y^* + W_2 + hU_2)(I_m + YY^*)^{-1} - A \right\|_F^2 \\
&\quad - \left\| (W_1 Y^* + W_2)(I_m + YY^*)^{-1} - A \right\|_F^2 \\
&= \left\| h(U_1 Y^* + U_2)(I_m + YY^*)^{-1} + (W_1 Y^* + W_2)(I_m + YY^*)^{-1} - A \right\|_F^2 \\
&\quad - \left\| (W_1 Y^* + W_2)(I_m + YY^*)^{-1} - A \right\|_F^2 \\
&= h^2 \left\| (U_1 Y^* + U_2)(I_m + YY^*)^{-1} \right\|_F^2 \\
&\quad + 2h \Re \left\langle (W_1 Y^* + W_2)(I_m + YY^*)^{-1} - A, (U_1 Y^* + U_2)(I_m + YY^*)^{-1} \right\rangle_F,
\end{aligned}
$$

from which we conclude that:

$$
\begin{aligned}
\nabla_U f(W_1, W_2) &= 2 \Re \left\langle (W_1 Y^* + W_2)(I_m + YY^*)^{-1} - A, (U_1 Y^* + U_2)(I_m + YY^*)^{-1} \right\rangle_F \\
&= 2 \Re \, trace(I_m + YY^*)^{-1}(YU_1^* + U_2^*)\left( (W_1 Y^* + W_2)(I_m + YY^*)^{-1} - A \right) \tag{26} \\
&= 2 \Re \, trace(YU_1^* + U_2^*)\left( (W_1 Y^* + W_2)(I_m + YY^*)^{-1} - A \right)(I_m + YY^*)^{-1}.
\end{aligned}
$$

Now, a necessary condition for $(W_1, W_2)$ to be a minimum point for $f$ is that $\nabla_U f(W_1, W_2) = 0$ for any $U = (U_1, U_2)$ as above. This implies that

$$\left( (W_1 Y^* + W_2)(I_m + YY^*)^{-1} - A \right)(I_m + YY^*)^{-1} = 0, \tag{27}$$

which is equivalent to

$$W_1 Y^* + W_2 - A(I_m + YY^*) = 0. \tag{28}$$

Similarly, for $g$, we have:

$$\nabla_U g(W_1, W_2) = \frac{\partial g}{\partial U}(W_1, W_2) = \lim_{h \to 0^+} \frac{g(W_1 + hU_1, W_2 + hU_2) - g(W_1, W_2)}{h}. \tag{29}$$

We compute:

$$
\begin{aligned}
g(W_1 + hU_1, W_2 + hU_2) - g(W_1, W_2) &= \left\| ((W_1 + hU_1)Y^* + W_2 + hU_2)(I_m + YY^*)^{-1}Y - B \right\|_F^2 \\
&\quad - \left\| (W_1 Y^* + W_2)(I_m + YY^*)^{-1}Y - B \right\|_F^2 \\
&= \left\| h(U_1 Y^* + U_2)(I_m + YY^*)^{-1}Y + (W_1 Y^* + W_2)(I_m + YY^*)^{-1}Y - B \right\|_F^2 \\
&\quad - \left\| (W_1 Y^* + W_2)(I_m + YY^*)^{-1}Y - B \right\|_F^2 \\
&= h^2 \left\| (U_1 Y^* + U_2)(I_m + YY^*)^{-1}Y \right\|_F^2 \\
&\quad + 2h \Re \left\langle (W_1 Y^* + W_2)(I_m + YY^*)^{-1}Y - B, (U_1 Y^* + U_2)(I_m + YY^*)^{-1}Y \right\rangle_F,
\end{aligned}
$$

from which we conclude that:

$$
\begin{aligned}
\nabla_U g(W_1, W_2) &= 2\Re\Big\langle (W_1 Y^* + W_2)(I_m + YY^*)^{-1}Y - B, (U_1 Y^* + U_2)(I_m + YY^*)^{-1}Y \Big\rangle_F \\
&= 2\Re\, trace\, Y^*(I_m + YY^*)^{-1}(YU_1^* + U_2^*)\Big((W_1 Y^* + W_2)(I_m + YY^*)^{-1}Y - B\Big) \\
&= 2\Re\, trace(YU_1^* + U_2^*)\Big((W_1 Y^* + W_2)(I_m + YY^*)^{-1}Y - B\Big)Y^*(I_m + YY^*)^{-1}.
\end{aligned}
\tag{30}
$$

A necessary condition for $(W_1, W_2)$ to be a minimum point for $g$ is that $\nabla_U g(W_1, W_2) = 0$ for any $U = (U_1, U_2)$ as above. This implies that

$$
\Big((W_1 Y^* + W_2)(I_m + YY^*)^{-1}Y - B\Big)Y^*(I_m + YY^*)^{-1} = 0,
\tag{31}
$$

is equivalent to

$$
\begin{aligned}
&\Big((W_1 Y^* + W_2)(I_m + YY^*)^{-1}Y - B\Big)Y^* = 0 \\
&\leftrightarrow (W_1 Y^* + W_2)YY^*(I_m + YY^*)^{-1} - BY^* = 0 \\
&\leftrightarrow (W_1 Y^* + W_2)YY^* - BY^*(I_m + YY^*) = 0.
\end{aligned}
\tag{32}
$$

Gathering (28) and (32), we obtain:

$$
\begin{bmatrix} W_2 & W_1 \end{bmatrix}\begin{bmatrix} I_m & YY^* \\ Y^* & Y^*YY^* \end{bmatrix} = \begin{bmatrix} A(I_m + YY^*) & BY^*(I_m + YY^*) \end{bmatrix},
\tag{33}
$$

which, in view of Lemma 1, implies that the best approximation in terms of the Frobenius norm is:

$$
\begin{bmatrix} W_2 & W_1 \end{bmatrix} = \begin{bmatrix} A(I_m + YY^*) & BY^*(I_m + YY^*) \end{bmatrix}\begin{bmatrix} I_m & YY^* \\ Y^* & Y^*YY^* \end{bmatrix}^{+}.
\tag{34}
$$

Finally, we define the projection $P_{\mathcal{C}_1}$ by $(A, B) \mapsto \left(\widehat{A}, \widehat{B}\right)$, where:

$$
\begin{cases}
\widehat{A} = (W_1 Y^* + W_2)(I_m + YY^*)^{-1} \\
\widehat{B} = (W_1 Y^* + W_2)(I_m + YY^*)^{-1}Y,
\end{cases}
\tag{35}
$$

where $W_1, W_2$ are given by (34). The projection $P_{\mathcal{C}_1}$ is realized in Algorithm 3.

Algorithm 1, as presented below, which we call the Alternating Projections Hard Successive Interference Cancellation (AP-HSIC), solves the problem of finding $\Delta H$ and $S$, such that $(H + \Delta H)^+ Y = S$, where $S \in \mathbb{B}\left(0, \sqrt{2nk}\right)$, after which it uses $P_{QPSK}$ (or the related modulation projection) in order to project $S$ into $QPSK^{n \times k}$ as the final decision. In its main loop, it computes $\left((H + \Delta H_{t+1})^+, S_{t+1}\right)$ by applying $P_{\mathcal{C}_1}$ to $\left((H + \Delta H_t)^+, S_t\right)$ and next, by applying $P_{\mathcal{C}_0}$ to the result, thus alternating between $\mathcal{C}_1$ and $\mathcal{C}_0$ until convergence to an intersection point (i.e., in $\mathcal{C}_1 \cap \mathcal{C}_0$) is verified.

We can start from any couple; however, a good choice is needed in order to obtain faster convergence, which we took as $\Delta H_0 = 0$, $S_0 = H^+ Y$ with the last $H$ known to the receiver, before the jamming attack began, which is a good starting point, as revealed by Theorem A3.

**Remark 2.** *Note that $Y = (H + \Delta H)S$ with the true $S$ implies that*

$$
\begin{aligned}
(H + \Delta H)(H + \Delta H)^+ Y &= (H + \Delta H)(H + \Delta H)^+(H + \Delta H)S \\
&= (H + \Delta H)S = Y.
\end{aligned}
$$

*Let $\widetilde{S} = (H + \Delta H)^+ Y$ be computed by Algorithm 1. Then,*

$$
(H + \Delta H)\widetilde{S} = (H + \Delta H)(H + \Delta H)^+ Y = Y.
$$

*Therefore, by solving $(H + \Delta H)^+ Y = S$ instead of $Y = (H + \Delta H)S$, we do not lose the generality.*

---

**Algorithm 2: For $P_{\mathcal{C}_0}$**

---

**Require:** *Matrix and Arithmetic Operations*
    **Input:** $(A, B)$ such that $A$ is $n \times m$ and $B$ is $n \times k$.
    **Output:** $\left( \widehat{A}, \widehat{B} \right) \in \mathcal{C}_0$ closest to $(A, B)$.

1. **if** $\|B\|_F > \sqrt{2nk}$ **then**
2.      $\beta \leftarrow \frac{\sqrt{2nk}}{\|B\|_F}$
3. **else**
4.      $\beta \leftarrow 1$
5. **end if**
6. $\left( \widehat{A}, \widehat{B} \right) \leftarrow (A, \beta B)$
7. **return** $\left( \widehat{A}, \widehat{B} \right)$

---

**Algorithm 3: For $P_{\mathcal{C}_1}$**

---

**Require:** *An algorithm for computing Moore–Penrose Pseudoinverses*
    **Input:** $(A, B)$ such that $A$ is $n \times m$ and $B$ is $n \times k$ and $Y$.
    **Output:** $\left( \widehat{A}, \widehat{B} \right) \in \mathcal{C}_1$ closest to $(A, B)$.

1. $\begin{bmatrix} W_2 & W_1 \end{bmatrix} \leftarrow \begin{bmatrix} A(I_m + YY^*) & BY^*(I_m + YY^*) \end{bmatrix} \begin{bmatrix} I_m & YY^* \\ Y^* & Y^*YY^* \end{bmatrix}^+$
2. $\widehat{A} \leftarrow (W_1 Y^* + W_2)(I_m + YY^*)^{-1}$
3. $\widehat{B} \leftarrow (W_1 Y^* + W_2)(I_m + YY^*)^{-1} Y$
4. **return** $\left( \widehat{A}, \widehat{B} \right)$

---

*Algorithm Correctness, Convergence, and Complexity*

    The general problem that we need to solve here is the problem $Y = (H + \Delta H)S$, for the unknowns $\Delta H$ and $S$, where $S \in QPSK^{n \times k}$. We assume that $H \in \mathbb{C}^{m \times n}$ is known to the receiver; however, this is not a mandatory assumption since we can rename $H + \Delta H$ as an (unknown) $H$ and solve the problem indifferently. The aforementioned general problem is non-convex since it involves the multiplication $\Delta H \cdot S$ of two unknown variables $\Delta H$ and $S$, and $QPSK^{n \times k}$ is a discrete set (with $4^{nk}$ elements) which is obviously non-convex. Indeed, the decision problem: "Given $H, \Delta H, Y$, does there exist $S \in QPSK^{n \times k}$ such that $Y = (H + \Delta H)S$?" is NP-hard to solve, as one can prove the existence of a polynomial-time reduction from the SUBSET-SUM problem. Therefore, solving the problem $Y = (H + \Delta H)S$ where $\Delta H$ is also unknown, is harder, and under the widespread belief that NP$\neq$P, it is not expected to have an efficient (i.e., polynomial-time) exact algorithm. Therefore, the proposed algorithm is an efficient algorithm as an approximation algorithm.

    In the previous section, we have shown that we may assume that $H + \Delta H$ is full-rank, thus leading to the relaxed problem: $(H + \Delta H)^+ Y = S$, where $S \in QPSK^{n \times k}$. The constraint $S \in QPSK^{n \times k}$ was replaced by a closed convex approximation envelope $S \in \mathbb{B}\left(0, \sqrt{2nk}\right)$ and the parametrization in terms of $W_1, W_2$ of all the solutions for $(H + \Delta H)^+ Y = S$ was given. Next, the closed convex sets $\mathcal{C}_0$ and $\mathcal{C}_1$ were defined, and their related projections (of an outer point to its closest point in the related set) $P_{\mathcal{C}_0}$ and $P_{\mathcal{C}_1}$ were defined. We, therefore, need an intersection point, i.e., a point in $\mathcal{C}_0 \cap \mathcal{C}_1$.

    For this purpose, we use the Alternating Projections (AP) algorithm, which is a known efficient algorithm that finds a point of intersection of a collection of closed convex sets, and was proven to have a linear rate of convergence globally (see [39,40]) and linear rate of convergence locally for non-convex sets (see [41]). Let $\epsilon_0 = \sqrt{\mathbb{E}\left[\|S_0 - S_*\|_F^2 + \|\Delta H_0 - \Delta H_*\|_F^2\right]}$ denote the initial error, where $\left((H + \Delta H_0)^+, S_0\right)$ is the initial point in the search space and $\left((H + \Delta H_*)^+, S_*\right)$ is a solution to the problem $Y = (H + \Delta H)S$ where $S_* \in QPSK^{n \times k}$, which, in view of Remark 2, is a point of the intersection $\mathcal{C}_0 \cap \mathcal{C}_1$. We may therefore take $\left((H + \Delta H_*)^+, S_*\right)$ to be the point of intersection $\mathcal{C}_0 \cap \mathcal{C}_1$ with $S_* \in QPSK^{n \times k}$ that is the closest to the initial point. Therefore, the algorithm converges to a point $\left((H + \Delta H)^+, S\right)$ of intersection $\mathcal{C}_0 \cap \mathcal{C}_1$, where $S \in \mathbb{B}\left(0, \sqrt{2nk}\right)$ is the closest to $S_* \in QPSK^{n \times k}$.

Let $\epsilon_t = \sqrt{\mathbb{E}\left[\|S_t - S_*\|_F^2 + \|\Delta H_t - \Delta H_*\|_F^2\right]}$ denote the error after iteration $t$ was executed. Then, the linearity of convergence means that $\lim_{t\to+\infty} \frac{\epsilon_t}{\epsilon_{t-1}} = \alpha$ for some $0 < \alpha < 1$. It follows that there exists $c > 0$ such that $\epsilon_t \leq c\epsilon_0 \alpha^t$, for any $t$. Therefore, for a given error threshold $\epsilon > 0$, $c\epsilon_0 \alpha^t \leq \epsilon$ implies that the needed number of iterations to achieve the goal is $t = \left\lceil \frac{1}{\ln(\alpha)} \left( \ln\left(\frac{\epsilon}{\epsilon_0}\right) - \ln(c) \right) \right\rceil$, that is $t_f = O\left(\left|\ln\left(\frac{\epsilon}{\epsilon_0}\right)\right|\right)$, where $t_f$ is the final number of iterations. The complexity of a single iteration is $O\left(\max(m,n,k)^3\right)$ due to matrix multiplications, matrix inversions, and pseudo-inversions. We conclude that the overall complexity of the proposed AP-HSIC algorithm is: $O\left(\max(m,n,k)^3 \left|\ln\left(\frac{\epsilon}{\epsilon_0}\right)\right|\right)$.

Since the only constraint on $\Delta H$ is that it should satisfy $(H + \Delta H)^+ Y = S$, and $\Delta H_t$ satisfy this constraint for any $t$, we may assume that $\Delta H_{t_f} = \Delta H_*$, where $t_f$ is the final iteration. Therefore, $\epsilon_{t_f} = \sqrt{\mathbb{E}\left[\left\|S_{t_f} - S_*\right\|_F^2\right]}$, and using Theorem A4, we conclude that:

$$\epsilon_{t_f} \leq \frac{n}{\sqrt{mSNR}}. \tag{36}$$

Or, in other words, if the given error threshold is $\epsilon$, then $\frac{n}{\sqrt{mSNR}} \leq \epsilon$ will achieve the goal $\epsilon_{t_f} \leq \epsilon$. In this case, the needed SNR is:

$$SNR \geq \frac{n^2}{m\epsilon^2}. \tag{37}$$

Note that the monotone convergence of the AP algorithm implies that $S_t$ is between $S_{t-1}$ and $S_*$, and $\Delta H_t$ is between $\Delta H_{t-1}$ and $\Delta H_*$. It follows that $\|S_t - S_{t-1}\|_F \leq \|S_{t-1} - S_*\|_F$ and $\|\Delta H_t - \Delta H_{t-1}\|_F \leq \|\Delta H_{t-1} - \Delta H_*\|_F$, from which we conclude that:

$$\mathbb{E}\left[\|S_t - S_{t-1}\|_F^2 + \|\Delta H_t - \Delta H_{t-1}\|_F^2\right]$$
$$\leq \mathbb{E}\left[\|S_{t-1} - S_*\|_F^2 + \|\Delta H_{t-1} - \Delta H_*\|_F^2\right] = \epsilon_{t-1}^2.$$

Therefore, if $\epsilon_{t-1}^2 \leq \epsilon^2$, then $\epsilon_t^2 \leq \epsilon^2$ and we can break the whole loop (Algorithm 1 line 7) since the AP algorithm has converged. On the other hand, if $\mathbb{E}\left[\|S_t - S_{t-1}\|_F^2 + \|\Delta H_t - \Delta H_{t-1}\|_F^2\right] > \epsilon^2$, then $\epsilon_{t-1}^2 > \epsilon^2$ means that the AP algorithm has not converged yet. This explains the condition in Algorithm 1 line 7.

Since we may assume that $m, n, k$ are bounded, therefore the size of the matrices, and specifically $\max(m,n,k)^3$, enter the complexity of the proposed AP-HSIC algorithm as a constant multiplicative factor. Thus, in this case, the complexity would be dominated by the relation between $\epsilon$ and $\epsilon_0$, which causes the algorithm to undergo several iterations that are shown in the following simulations to be decreasing as a function of the SNR at a rate that is proportional to $\frac{n}{\sqrt{mSNR}}$, thus assessing Theorem A4 and specifically (A6). In the simulations, we have fixed the error threshold to be $\epsilon = 10^{-9}$, where each simulation period was fixed to $200 \cdot T_s = 3.2$ msec and where the symbol-time was fixed to $T_s = 16$ μsec, which are custom values for the problem. In each simulation, the algorithm is shown to have no more than eight iterations (in the range $-5 \leq SNR \leq 25$ dB), where the convergence was achieved within the simulation period for all simulations.

If, in other scenarios or under different assumptions, the algorithm does not converge within the expected time period, it only means that more computational power is needed at the receiver side because, as noted earlier, the algorithm always converges (in a linear rate of convergence). Finally, note that the algorithm can work with other modulation orders, e.g., 16,32-QAM, since it only needs the definition of a relatively small closed convex set that contains all the possible symbol matrices (i.e., the modulation set) and a projection onto the modulation set that projects a general point onto the closest point in the modulation set.

## 4. Simulations and Numerical Results

To prove the superiority of the proposed AP-HSIC method, we present (not only on the MATLAB platform but also on the SIMULINK platform) simulations of a communication environment under all of the assumptions discussed in Section 2. Our experiments are based on developing two different simulations, based on the two architectures described in Section 2, and operating under the same conditions and in the same challenging scenarios. We compared the performance of these two simulations in graphical and numerical aspects. Each simulation presents a specific design consisting of several stages in which different disturbance scenarios are run.

To simplify the architectures, the MGSTC transmitter encoder, common to both architectures, includes three OSTBC components (in the transmitter and in the general interference). Each component is equipped with two transmission antennas, meaning that the transmitter and the interference each have six transmission antennas ($N_t = 6$). On the receiver side, we receive the signal $Y$ through six receiving antennas ($N_r = 6$). In addition to the estimated MIMO channel matrix, the received signal is streamed to the MGSTC decoding algorithm in the first architecture and into the AP-HSIC algorithm in the second architecture. The output of the algorithms comprises three Maximal-Ratio-Combiners (MRC) in each receiver and in both architectures for the final decoding, as can be seen in Figure 1.

The first simulation included an MGSTC transmitter based on an information generator block, symbol modulation block, and OSTBC-encoder blocks. The blocking scheme was described in Figure 1 and uses the SIMULINK blocks described in Figure 2. The MIMO channel is a quasi-static Flat-Rayleigh-Fading, which can change the AWGN intensity parameter (variation). The receiver blocks include an MGSTC decoding algorithm and MRC blocks. In addition, the scheme includes an interference simulation block combining two kinds of interference—PBN and general interference (same as the transmitter scheme we described)—each with a MIMO quasi-static Flat-Rayleigh-Fading. Finally, we included a BER calculator to measure the practical effect derived from the system's performance.

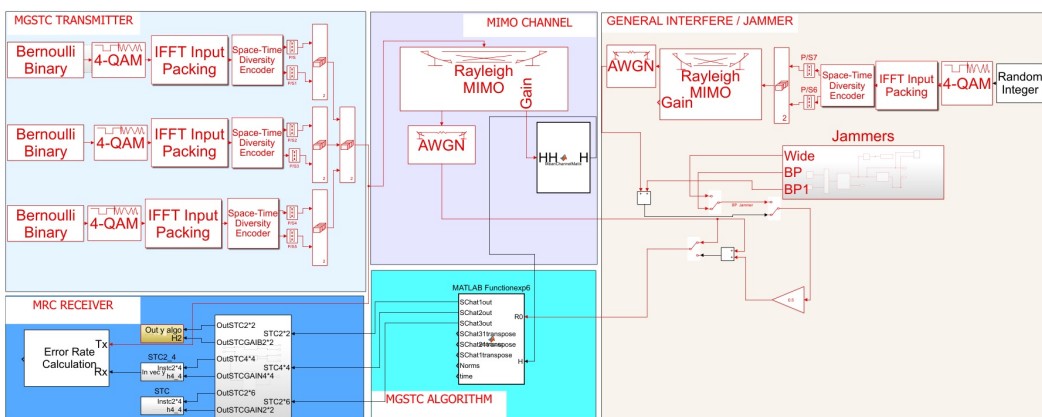

**Figure 2.** Scheme blocks of the simulation of the MGSTC algorithm based on the SIMULINK/MATLAB platform.

The second simulation was based on the same MGSTC transmitter and MIMO channel matrix block with the same interference block. However, on the receiver side, instead of an MGSTC decoding algorithm block, we changed the receiver front and strengthened the receiver ability with our proposed AP-HSIC block, which operates in a parallel decoding communication mode. The SIMULINK blocks of the second simulation are described in Figure 3.

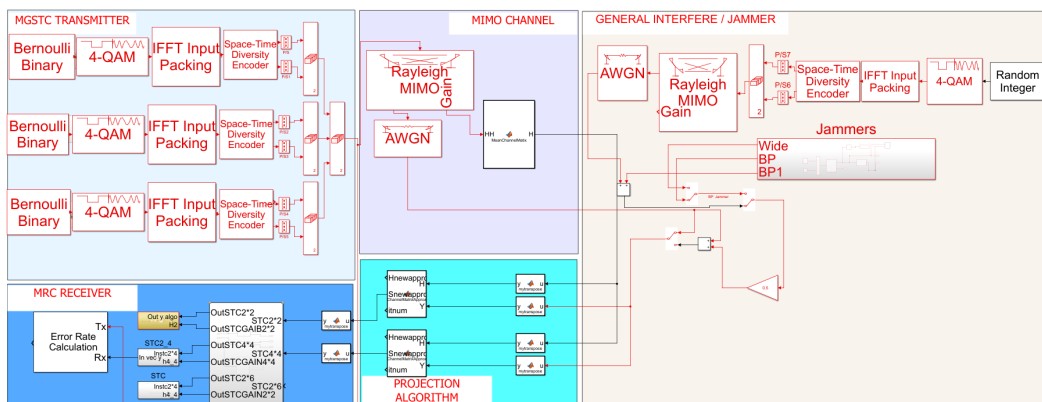

**Figure 3.** Scheme blocks of the simulation of the AP-HSIC based on the SIMULINK/MATLAB platform.

We examined the architectures of the systems in terms of BER vs. SNR/SIR, by running three interference scenarios separately—high-level AWGN, PBN, and general interference—under three modulation orders—4-, 8-, and 16-PSK—at the transmitter (and at the general interference transmitter).

The simulation performance parameters are based on [42]. As is well known, the systematic BER of the MGSTC decoding algorithm is always the BER of the last iteration. The systematic BER of the MGSTC decoding algorithm was compared with the average total BER of the AP-HSIC, as illustrated in the following Figures.

### 4.1. MGSTC and AP-HSIC Simulations under High-Level AWGN

As described previously in this section, in the first scenario, we simulated the case of high-level AWGN based on QPSK, 8–PSK, and 16–PSK modulations. In Figure 4a, we illustrate the performance graph based on the QPSK modulation of BER vs. SNR. It should be noted that the performance of the AP-HSIC (purple line) was significantly superior compared to that of the MGSTC decoding algorithm (yellow line, which relates to its systematic performance). The superiority of the AP-HSIC performance has also been proven when compared to the two iterations of the MGSTC decoding algorithm (equivalent to a 2 × 2 MIMO channel—the blue line; and equivalent to a 2 × 4 MIMO channel—the orange line) under the interference mentioned above. Note that the transmission power in the 2 × 2 and in the 2 × 4 MGSTC decoding algorithm was higher than that in the 6 × 6 AP-HSIC scheme for the fairness of comparison considerations.

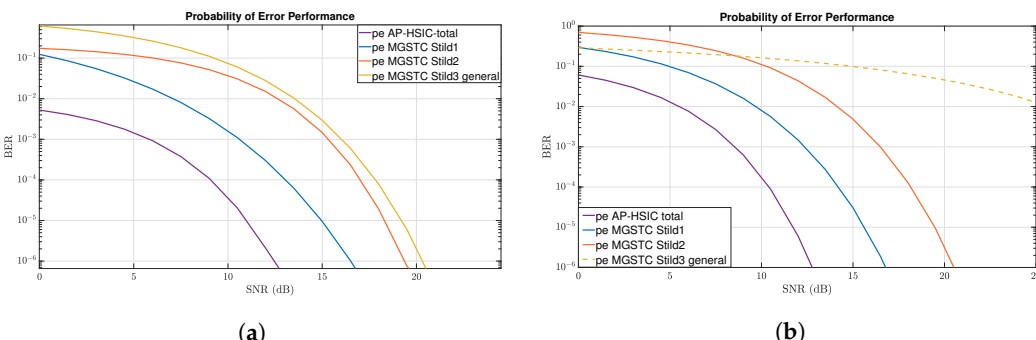

(**a**)          (**b**)

**Figure 4.** BER vs. SNR of MGSTC and AP-HSIC. (**a**) Comparison under QPSK modulation in the presence of high-level AWGN. (**b**) Comparison under 8-PSK modulation in the presence of high-level AWGN.

The graphical analysis of BER vs. SNR consists of three main characteristics that quantify the quality of the system's performance. The first is the ability of the specific system to achieve the lowest BER at a low SNR. The following illustrations show that the AP-HSIC achieved the lowest BER in the low SNR range in all the test cases. The second is the difference in the transmission power level, which must be invested to converge to a certain BER threshold. For example, comparing the simulation of the MGSTC decoding algorithm to the AP-HSIC under high-leveled AWGN (Figure 4a) at $BER = 10^{-6}$ shows that the AP-HSIC achieved an advantage of ~8 dB in terms of SNR. The

third—and the most significant—difference is the level of diversity gain between the systems. It can be seen from Figure 4a that the AP-HSIC achieved a much higher diversity gain. In the following section, we mathematically prove this issue.

Another aspect relates to the cumulative BER in the serial MIMO-mode decoding process based on the MGSTC decoding algorithm. As we described in Section 2, the re-evaluation accuracy of a current iteration totally depends on the previous iteration. From all of the BER vs. SNR graphs, it can be seen that the BER threshold obtained in the first iteration of the MGSTC decoding algorithm (blue line) was accumulated and dragged. This caused a more significant BER threshold in the second iteration (orange line), which became the worst in the third iteration (yellow line), causing a destructive chain reaction leading to the deterioration of the modulation order and increasing the transmission power (which was not even effective in the 8-PSK-black dashed line or 16-PSK-green dashed line cases, as can be seen from Figure 5b). In contrast, due to the parallel decoding process that the AP-HSIC is based on, the BER of the AP-HSIC in all the decoding processes remained consistent and robust. In Figure 4b, we can see other experimental results under the same conditions as those depicted in Figure 4a, but with a different modulation order (8-PSK) at the MGSTC transmitter. The insights from the results of these simulations prove that the AP-HSIC had a better performance when compared to all three MGSTC iterations.

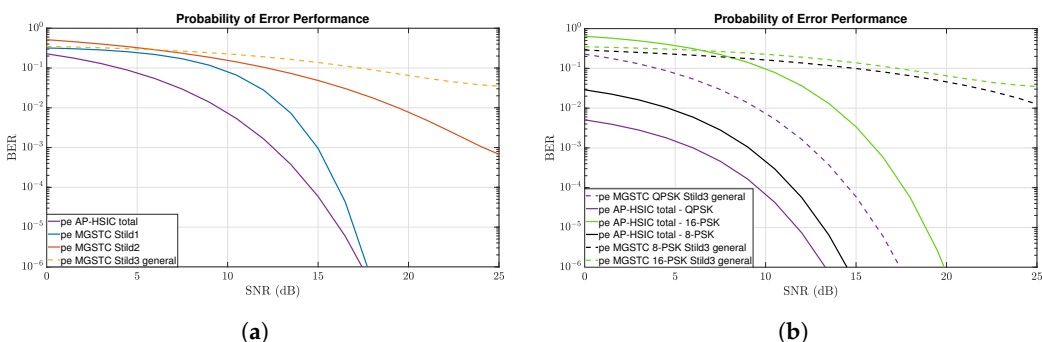

**Figure 5.** BER vs. SNR of MGSTC and AP-HSIC. (**a**) Comparison under 16-PSK in the presence of high-leveled AWGN. (**b**) Comparisons using QPSK, 8-PSK,16-PSK, in the presence of high-level AWGN.

Moreover, we can see that, in noisy environments, the reduction in the MGSTC transmission power in the spatial paths of the third sub-system is reflected in Figure 4b, which is the systematic graph of the BER of the whole MGSTC system. Note that the system has a reciprocal relationship between the diversity gain and the transmission power per symbol for every antenna in the transmitter ($\frac{E_S}{4}$). This weakness is reflected in the fact that, under the simulation with 8-PSK, the graph (Figure 4b) does not converge to any value that can be considered as a threshold value for producing any reliable information.

Figure 5a presents the simulation results with the 16-PSK modulation order under high-leveled AWGN. In this situation, the performance graph of the MGSTC decoding algorithm was even worse than in the previous scenarios. In addition to the graph of the third iteration (yellow dashed line), the graph of the second iteration (orange line) also did not converge towards a BER production threshold value. Thus, in this situation, only the first iteration of the MGSTC decoding algorithm (blue line) produced effective links (i.e., effectively, the system became a $2 \times 2$ MIMO); while in the AP-HSIC (purple line), the full potential of the MIMO-parallel communication-mode ($6 \times 6$ MIMO) is reflected. The illustrations summarizing all the BER vs. SNR graphs of the MGSTC decoding and the AP-HSIC on QPSK, 8-PSK, and 16-PSK modulations in the presence of high-leveled AWGN are presented in Figure 5b. This figure shows that the systems reinforced with the AP-HSIC are the only systems to converge to low values of BER at a given rate of information and a given SNR, under the above modulation orders, in this challenging scenario. In Figure 6a, under the QPSK case, we show the number of internal iterations required by the AP-HSIC to converge. We can also see how the rate of convergence depends on the SNR.

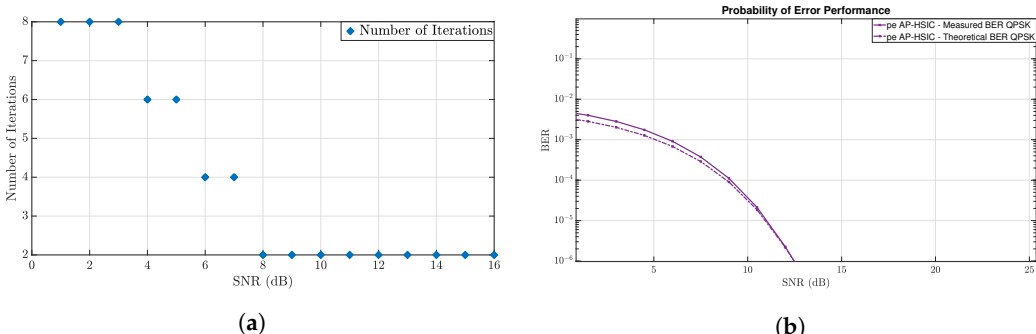

**Figure 6.** Number of iterations of the AP-HSIC and measured-theoretical AP-HSIC QPSK BER: (**a**) Number of iterations of the AP-HSIC vs. SNR under QPSK modulation; and (**b**) Comparisons between the measured-theoretical AP-HSIC QPSK BER.

### *4.2. MGSTC Algorithm and AP-HSIC under PBN Simulation*

As was presented at the beginning of this section, the PBN interference is characterized by a noise energy concentration located in the bandwidth part of the received signal. In our simulations, the ratio between the interference bandwidth and the received signal bandwidth was 0.5. In Figure 7a, we present a performance comparison of the two considered architectures under PBN interference and QPSK modulation order at the transmitter. Note that the performance of the AP-HSIC indicated that for SNR below $\approx 4$ dB, it achieved $BER = 10^{-6}$, while the MGSTC decoding algorithm obtained the same BER with SNR below $\approx 14$ dB, demonstrating the efficacy of $\approx 10$ dB in favor of the AP-HSIC.

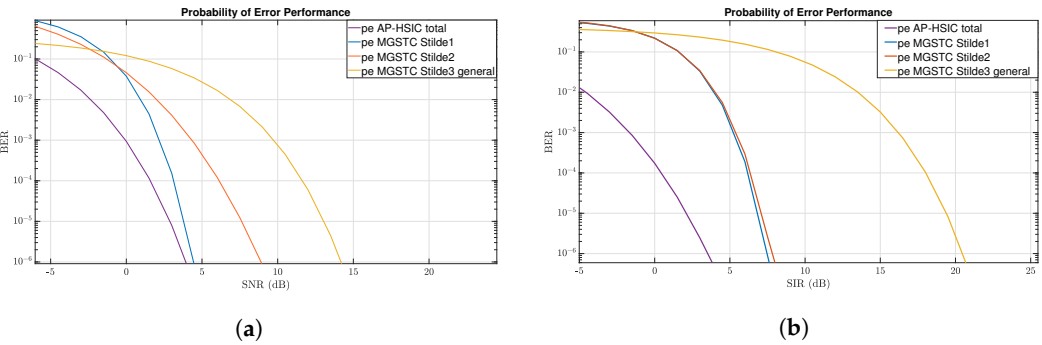

**Figure 7.** BER vs. SNR of MGSTC and AP-HSIC. (**a**) Comparison under QPSK modulation in the presence of PBN interference. (**b**) Comparison under QPSK modulation in the presence of general interference.

### *4.3. MGSTC Algorithm and AP-HSIC under General Interference Scenarios*

The most destructive of the three scenarios is the general interference one. From Figure 7b, we can draw several important conclusions. The first is based on analyzing the system performance graph in its negative and lower-positive regions of SIR. This analysis leads to the conclusion that, in the case of the AP-HSIC, there was a decrease in the BER, depending on the SIR, compared to the MGSTC decoding algorithm, while the graph of the MGSTC decoding algorithm remained almost constant until a relatively high SIR. Another insight is the system's ability to overcome an interference while maintaining maximum energy efficiency. Thus, from Figure 7b, we can see that with the reinforcement of the AP-HSIC-self-computational feedback—compared to methods based on spatial selectivities, such as the MGSTC decoding algorithm—it is possible to save up to SIR $\approx 17$ dB at $BER = 10^{-6}$.

## 5. Discussion, Conclusions, and Future Directions

It is undisputed that resolving interfering offset must be based on spatial selectivity, accuracy in the channel estimation process, and channel sharing with creating feedback between the receiver and the transmitter (and vice versa). However, to produce high-interference cancellation capabilities and enable cognitive and collaborative radio communications, the completion of an intelligent and efficient algorithm reinforcing the receiver is required. Such computational feedback can handle a variety of interference scenarios, as reflected in the simulation results, and improve the decoding process and receiver capabilities. The creation of advanced self-computational feedback capabilities

on the receiver is critical for developing the expected MIMO modern wireless communications and advancing the development of MIMO techniques. These technologies require the existence of Ultra-Reliable Low-Latency Communication Coding (URLLC) [43], some of which must be able to respond immediately, such as robots, autonomous tools, vehicles, and medical equipment. Moreover, some technologies physically do not allow feedback between the receiver and the transmitter (e.g., satellite communications), so self-computational feedback is critical when applying these technologies.

The simulation results presented herein demonstrated the required trade-off between energy efficiency and computational capabilities. The results obtained showed that, in cases where there is the requirement to increase the volume of information and transmit at a higher data rate (i.e., in situations where a higher modulation order is required), in addition to the system being under different interference scenarios or jamming attacks, investing in transmission power only does not necessarily lower the BER. Equally important, another insight in analyzing these results was found in the analysis of negative or relatively low-positive SIR regions, where the AP-HSIC significantly lowered the BER compared to the MGSTC technique.

In future research, we plan to expand the concept of computational feedback, deal with the assumptions of selective channels in frequency and time–space, and deal with non-stationary channels in the broadest sense.

**Author Contributions:** Methodology, Investigation, A.E.; Supervision, Investigation Y.P. (Yossi Peretz), and Y.P. (Yosef Pinhasi); Conceptualization, Methodology, Formal analysis, Software Y.P. (Yossi Peretz). All authors have read and agreed to the published version of the manuscript.

**Funding:** This research received no external funding.

**Conflicts of Interest:** The authors declare no conflict of interest.

**Appendix A. Theorems Proofs**

**Theorem A1.** *The set of full-rank matrices is dense in the set of all matrices, i.e., for any $A \in \mathbb{C}^{m \times n}$ and any $\epsilon > 0$, there exists a full-rank matrix $A_\epsilon$ such that $\|A - A_\epsilon\|_F \leq \epsilon$.*

**Proof.** Assume w.l.o.g. that $n \leq m$ and that $\ell = rank(A) \leq n$. If $\ell = n$, there is nothing to prove. Otherwise, let an S.V.D. of $A$ be given by $A = U\Sigma V^*$, where $U$ is an $m \times m$ unitary matrix, $V$ is an $n \times n$ unitary matrix, and $\Sigma$ is an $m \times n$ rectangular diagonal matrix with entries $\sigma_1 \geq \sigma_2 \geq \ldots \geq \sigma_\ell > 0$. Let $\sigma_k = \frac{\epsilon}{\sqrt{n-\ell}}$ for $k = \ell + 1, \ldots, n$, let $\Sigma_\epsilon$ be the $m \times n$ rectangular diagonal matrix with main-diagonal entries $\sigma_1, \ldots, \sigma_n$ in this order. Finally, let $A_\epsilon = U\Sigma_\epsilon V^*$. Then, $rank(A_\epsilon) = n$ and

$$
\begin{aligned}
\|A - A_\epsilon\|_F^2 &= \|U(\Sigma - \Sigma_\epsilon)V^*\|_F^2 \\
&= \|\Sigma - \Sigma_\epsilon\|_F^2 \\
&= \sum_{k=\ell+1}^{n} \sigma_k^2 = \epsilon^2.
\end{aligned}
$$

This completes the proof.  □

**Theorem A2.** *Assume that the communication is made with some modulation not containing 0 and let $r_0$ be the radius of the modulation point with minimal radius. Assume that $k \geq \max(m, n)$ and let $S = \begin{bmatrix} S_1 & S_2 & \cdots & S_\ell & S_{\ell+1} \end{bmatrix}$ be an OSTBC symbol matrix of signals, where $k = n\ell + q$, each $S_j$ is $n \times n$, chosen such that $S_j S_j^* \geq r_0^2 I_n$ for $j = 1, \ldots, \ell$, and $S_{\ell+1}$ is an arbitrary $n \times q$ matrix. In addition, also assume that $S$ is known to the receiver. Let $H_0 = Y S^+$. Then, $H_0$ is a random matrix such that:*

$$
\mathbb{E}\left[\|H - H_0\|_F^2\right] = \frac{\sigma^2}{k}\|S^+\|_F^2. \tag{A1}
$$

*Moreover, since $\|S^+\|_F$ is bounded above, by taking $k \to +\infty$, it follows that $\mathbb{E}\left[\|H - H_0\|_F^2\right] = 0$, implying that $H_0 = H$ A.S. holds.*

**Proof.** Note that $S$ has $rank(S) = n$. Therefore $S^+ = S^*(SS^*)^{-1}$ and $R_S = I_n - SS^+ = 0$. Now, $SS^* = \sum_{j=1}^{\ell+1} S_j S_j^* \geq \sum_{j=1}^{\ell} S_j S_j^* \geq \ell r_0^2 I_n$, implying that $(SS^*)^{-1} \leq \frac{1}{\ell r_0^2} I_n$. Therefore, $S^{+*}S^+ = (SS^*)^{-1}SS^*(SS^*)^{-1} = (SS^*)^{-1} \leq \frac{1}{\ell r_0^2} I_n$, from which we conclude that $\|S^+\|_F^2 = trace(S^{+*}S^+) \leq$

$\frac{n}{\ell r_0^2} = \frac{n^2}{(k-q)r_0^2}$. Now, we decompose $H = HSS^+ + HR_S = (Y - Z)S^+ + HR_S = H_0 - ZS^+ + HR_S = H_0 - ZS^+$ and obtain $\|H - H_0\|_F^2 = \|-ZS^+\|_F^2$. Therefore,

$$\mathbb{E}\left[\|H - H_0\|_F^2\right] = \mathbb{E}\left[\|-ZS^+\|_F^2\right] = \mathbb{E}[trace(S^{+*}Z^*ZS^+)]$$
$$= trace(S^{+*}\mathbb{E}[Z^*Z]S^+) = trace\left(S^{+*}\left(\frac{\sigma}{k}I_k\right)trace\left(\frac{\sigma}{m}I_m\right)S^+\right)$$
$$= \frac{\sigma^2}{k}trace(S^{+*}S^+) = \frac{\sigma^2}{k}trace(S^{+*}S^+) = \frac{\sigma^2}{k}\|S^+\|_F^2$$
$$\leq \frac{\sigma^2 n^2}{k(k-q)r_0^2}.$$

Now, letting $k \to +\infty$, we conclude that $\mathbb{E}\left[\|H - H_0\|_F^2\right] = 0$, implying that $H_0 = H$ A.S. Note also that $H_0 = H$ A.S. immediately implies that $\mathbb{E}[H] = \mathbb{E}[H_0]$. $\square$

**Theorem A3.** *Let $H_0$ be an $m \times n$ matrix as in Theorem A2, and further assume that $m \geq n$ and that $\mathbb{E}[H_0]$ is full-rank. Let $S$ denote an unknown signal matrix sent by the transmitter, and let $Y = HS + Z$ be the signal measured by the receiver. Let $\widetilde{S} = \mathbb{E}[H_0]^+ Y = \mathbb{E}[H_0]^+ (HS + Z)$ (note that $\widetilde{S}$ is a random variable matrix, while $S$ is not random). Then,*

$$\mathbb{E}\left[\widetilde{S}\right] = S. \tag{A2}$$

**Proof.** Using the proof of Theorem A2, we have $\mathbb{E}[H] = \mathbb{E}[H_0]$. Since $m \geq n$ and $\mathbb{E}[H_0]$ is full-rank, we obtain that $\mathbb{E}[H_0]^+ \mathbb{E}[H_0] = I_n$. Now, we compute:

$$\mathbb{E}\left[\widetilde{S}\right] = \mathbb{E}\left[\mathbb{E}[H_0]^+ Y\right]$$
$$= \mathbb{E}[H_0]^+ \mathbb{E}[HS + Z]$$
$$= \mathbb{E}[H_0]^+ (\mathbb{E}[H]S + \mathbb{E}[Z])$$
$$= \mathbb{E}[H_0]^+ (\mathbb{E}[H]S + 0_{m \times k})$$
$$= \mathbb{E}[H_0]^+ \mathbb{E}[H_0]S = S.$$

$\square$

**Theorem A4.** *Let $Y = HS + Z$, where $H$ can be the ideal channel matrix or $H + \Delta H$, the resulting channel matrix of the disrupted channel. Assume that the exact $H$ (or the exact $H + \Delta H$, respectively) is known to the receiver and assume that $H$ (or $H + \Delta H$, respectively) is constant for the current session. Let $S$ be the true symbol matrix for the current session, and let $\widetilde{S} = H^+ Y$ be the approximated symbol matrix. Then, the distribution of $\widetilde{S} - S$ (given the constant matrix $H$ or $H + \Delta H$, respectively) is:*

$$\widetilde{S} - S \sim \mathcal{CMN}_{n \times k}\left(0_{n \times k}, \frac{\sigma}{m}H^+ H^{+*}, \frac{\sigma}{k}I\right). \tag{A3}$$

*Moreover, the expectation of the square error satisfies:*

$$\mathbb{E}\left[\left\|\widetilde{S} - S\right\|_F^2\right] = \frac{\|H\|_F^2 \|H^+\|_F^2}{mSNR} = \frac{P\|H^+\|_F^2}{mSNR}. \tag{A4}$$

**Proof.** Let us write $Y = HS + Z$ as $Y = \begin{bmatrix} H & I_m \end{bmatrix}\begin{bmatrix} S \\ Z \end{bmatrix}$. This implies that the minimal Frobenius-norm solution for $S$ and $Z$ is given by:

$\begin{bmatrix} H & I_m \end{bmatrix}^+ Y = \begin{bmatrix} S \\ Z \end{bmatrix}$, that is, $S = H^*(I_m + HH^*)^{-1}Y$, $Z = (I_m + HH^*)^{-1}Y$. The substitution of $Y = (I_m + HH^*)Z$ into $\widetilde{S} = H^+ Y$ yields $\widetilde{S} = H^+(I_m + HH^*)Z$. Now, $H^+ = \lim_{\epsilon \to 0^+} H^*(\epsilon I_m + HH^*)^{-1}$. Let $\widetilde{S}_\epsilon = H^*(\epsilon I_m + HH^*)^{-1}(I_m + HH^*)Z$. Then,

$$\widetilde{S}_\epsilon = H^*(\epsilon I_m + HH^*)^{-1}(I_m + HH^*)Z = H^*(\epsilon I_m + HH^*)^{-1}(\epsilon I_m + HH^* + (1 - \epsilon)I_m)Z$$
$$= H^*Z + (1 - \epsilon)H^*(\epsilon I_m + HH^*)^{-1}Z = S + (1 - \epsilon)H^*(\epsilon I_m + HH^*)^{-1}Z,$$

implying that: $\widetilde{S} = \lim_{\epsilon \to 0^+} \widetilde{S}_\epsilon = S + H^+ Z$. Assuming that $Z \sim \mathcal{CMN}_{n \times k}\left(0_{n \times k}, \frac{\sigma}{m} I_m, \frac{\sigma}{k} I\right)$, we conclude (A3). The latter implies that $\mathbb{E}\left[\left\|\widetilde{S} - S\right\|_F^2\right] = trace\left(\frac{\sigma}{m} H^+ H^{+*}\right) trace\left(\frac{\sigma}{k} I_k\right) =$
$= \frac{\sigma^2}{m}\left\|H^+\right\|_F^2$.

Let $r = rank(H)$, and let $H = \sum_{\ell=1}^r \sigma_\ell u_\ell v_\ell^*$ be an SVD of $H$. Then, $H^+ = \sum_{\ell=1}^r \frac{1}{\sigma_\ell} v_\ell u_\ell^*$. We also have $\|H\|_F^2 = \sum_{\ell=1}^r \sigma_\ell^2$ and $\left\|H^+\right\|_F^2 = \sum_{\ell=1}^r \frac{1}{\sigma_\ell^2}$. Finally, as $SNR = \rho = \frac{\|H\|_F^2}{\sigma^2} = \frac{P}{\sigma^2}$, we conclude that

$$\mathbb{E}\left[\left\|\widetilde{S} - S\right\|_F^2\right] = \frac{P\|H^+\|_F^2}{mSNR} = \frac{\|H\|_F^2 \|H^+\|_F^2}{mSNR} = \frac{\left(\sum_{\ell=1}^r \sigma_\ell^2\right)\left(\sum_{\ell=1}^r \frac{1}{\sigma_\ell^2}\right)}{mSNR}, \text{ proving (A4).}$$

Note that, in full-capacity mode, i.e., $\sigma_1 = \cdots = \sigma_\ell$, we have

$$\mathbb{E}\left[\left\|\widetilde{S} - S\right\|_F^2\right] = \frac{r^2}{mSNR} \leq \frac{n^2}{mSNR}, \tag{A5}$$

where $r = rank(H) \leq n$, since $n \leq m$. Furthermore, under the assumption that $H$ is full-rank, we have $r = rank(H) = n$. In that case,

$$\mathbb{E}\left[\left\|\widetilde{S} - S\right\|_F^2\right] = \frac{n^2}{mSNR}. \tag{A6}$$

$\square$

## Appendix B. Comparison between Theoretical Results and Simulations

Given Theorem A4, and specifically given (A6), let us relate the expected number of defected bits to the expected decoding error $\sqrt{\mathbb{E}\left[\left\|\widetilde{S} - S\right\|_F^2\right]}$ as:

$$\gamma \frac{n}{\sqrt{mSNR}}, \tag{A7}$$

where $\gamma$ is an assumed universal correction factor that depends on the modulation only.

Then,

$$BER = \frac{\gamma \frac{n}{\sqrt{mSNR}}}{2nk} = \frac{\gamma}{2k\sqrt{mSNR}}. \tag{A8}$$

Therefore, for the MGSTC simulated system, where $k = 402$ symbols, we have:

$$BER_{MGSTC} = \frac{\gamma}{2 \cdot 402 \cdot \sqrt{2 \cdot SNR}} + \frac{\gamma}{2 \cdot 402 \cdot \sqrt{4 \cdot \frac{SNR}{2}}} + \frac{\gamma}{2 \cdot 402 \cdot \sqrt{6 \cdot \frac{SNR}{4}}}$$
$$= \frac{\gamma}{804\sqrt{SNR}} \cdot \left(\sqrt{2} + \sqrt{2/3}\right). \tag{A9}$$

For the AP-HSIC, for fairness of comparison, we took its SNR per-antenna to be $\frac{2(1+1/2+1/4)}{6} SNR$, relative to the MGSTC-decoding algorithm. Therefore, we have:

$$BER_{PA-HSIC} = \frac{\gamma}{2 \cdot 402 \cdot \sqrt{6 \cdot \frac{7SNR}{12}}} = \frac{\gamma}{804\sqrt{SNR}} \cdot \sqrt{\frac{2}{7}} \tag{A10}$$

Taking the ratio, we obtain the following:

$$\frac{BER_{PA-HSIC}}{BER_{MGSTC}} = \frac{\sqrt{\frac{2}{7}}}{\sqrt{2} + \sqrt{2/3}} \approx 0.2396. \tag{A11}$$

To theoretically establish the simulation results, two approaches are represented, based on Formula (A8). The first approach is as follows. Taking the logarithm of (A8), we obtain:

$$10\log_{10}(BER) = 10\log_{10}(\gamma) + 10\log_{10}(n) - 5\log_{10}(m) - 5\log_{10}(SNR)$$
$$\sim_{SNR \to \infty} 10\log_{10}(\gamma) - 5\log_{10}(SNR). \tag{A12}$$

From this, we can conclude that

$$10 \log_{10}\left(\frac{BER_{PA-HSIC}}{BER_{MGSTC}}\right) \sim_{SNR\to\infty} -5 \log_{10}(SNR_{PA-HSIC}) + 5 \log_{10}(SNR_{MGSTC}), \quad (A13)$$

which, in view of (A7), should give us:

$$-5 \log_{10}(SNR_{PA-HSIC}) + 5 \log_{10}(SNR_{MGSTC}) \approx 10 \log_{10}(0.2396) \approx -6.2051 \text{ dB}. \quad (A14)$$

Indeed, looking at Figure 4a at $BER = 1 \cdot 10^{-5}$, we can see that the difference between the $SNR$ values was approximately 6 dB. The second approach is based on the following stages. First, we measure several different samples, with the AP-HSIC simulator (for the high-leveled AWGN, QPSK modulation order scenario), of $\sqrt{\mathbb{E}\left[\|\tilde{S} - S\|_F^2\right]}$ and the number of incorrect bits per-frame. The sampled values of $\sqrt{\mathbb{E}\left[\|\tilde{S} - S\|_F^2\right]}$ are defined as a vector $x$, and the number of incorrect bits per-frame as another vector $y$. In the second stage, we perform the calculation for the assumed universal correction factor $\gamma$, as $\gamma = \frac{2nk \sum_i x_i y_i}{\sum_i x_i^2}$, to minimize the mean-square-error of the linear approximation for it. In our calculation, we found $\gamma_{QPSK} = 2.53 \cdot 10^{-3}$. The last stage involves calculating the theoretical $i^{\text{th}}$ BER as $BER_i = \gamma_{QPSK} \frac{x_i}{2nk}$. In our simulations, we had $\sqrt{\mathbb{E}\left[\|\tilde{S} - S\|_F^2\right]} = 19$ under $SNR \approx 10.5$ dB. Thus, with $\gamma_{QPSK} = 2.53 \cdot 10^{-3}$, the theoretical BER was $\approx 1 \cdot 10^{-5}$, which is identical to the BER value shown in Figure 4a. A comparison between the measured AP-HSIC QPSK BER and the theoretical AP-HSIC QPSK BER is in Figure 4b.

Under the same simulation conditions but with an 8-PSK modulation order, the same theoretical calculation of the BER was carried out. In this case, we found $\gamma_{8-PSK} = 1.2$, and the theoretical BER under $SNR \approx 10.5$ dB was $1.04 \cdot 10^{-4}$. In comparison, the BER under the value of $SNR \approx 10.5$ dB in Figure 4b was $\approx 1 \cdot 10^{-4}$.

For the 16-PSK case, we again carried out the same calculation processes and found $\gamma_{16-PSK} = 6.1$, while the theoretical BER under $SNR \approx 10.5$ dB was $8.3 \cdot 10^{-3}$. The BER under the value of $SNR \approx 10.5$ dB in Figure 5a was $\approx 9 \cdot 10^{-3}$.

**Table A1.** Units and parameters used in the conducted simulations.

| Symbol | Parameter | Value/Description |
|---|---|---|
| $c_i$ | Number of OSTBC component encoders in the MGSTC transmitter | $i = 1, 2, 3$ |
| $n_i$ | Number of transmission antennas for every OSTBC component | $n_1 = 2, n_2 = 2$ $n_3 = 2$ |
| $N_t$ | Total number of transmission antennas in the MGSTC transmitter | 6 |
| $N_r$ | Sum of the total receiver antennas in the receiver | 6 |
| $S_{ci} = \begin{bmatrix} \vec{s}_{ci,1} \\ \vec{s}_{ci,2} \end{bmatrix}$ | Row vectors of block transmission symbol matrix | $\vec{s}_{ci,1}, \vec{s}_{ci,2}\ i = 1, 2, 3$ |
| $\vec{s}_{ci,1}$ | Transmission Alamouti coding operation in the first antenna of the OSTBC encoder | $t \to s_1(2j-1) = a$ $t + T_s \to s_1(2j) = -b^*$ $j = 1, \ldots, 201$ |
| $\vec{s}_{ci,2}$ | Transmission Alamouti coding operation in the second antenna of the OSTBC encoder | $t \to s_2(2j-1) = b$ $t + T_s \to s_2(2j) = a^*$ $j = 1, \ldots, 201$ |
| $T_{S_{ci}}$ | Sample time per frame | $16 \cdot 10^{-6}$ (s) |
| $R_b$ | Transmission bit rate | $174.825 \cdot 10^{-6}$ (bps) |
| $H_{TR}$ | Channel matrix response between the transmitter and the receiver | $[6 \times 6]$ |

**Table A2.** Units and parameters used in the conducted simulations.

| Symbol | Parameter | Value/Description |
|---|---|---|
| $H_{JR}$ | Channel matrix response between the general interference/jammer to the receiver | $[6 \times 6]$ |
| $N_j$ | Sum of the total transmission antennas in the MGSTC general interference | 6 |
| $S_J$ | The OSTBC matrix of the interferer | |
| $J_{ci} = \begin{bmatrix} \vec{j}_{ci,1} \\ \vec{j}_{ci,2} \end{bmatrix}$ | Row vectors of block transmission of the general interference/jammer | $\vec{j}_{ci,1}, \vec{j}_{ci,2}\ i = 1, 2, 3$ |
| $\vec{j}_{ci,1}$ | Transmission Alamouti coding operation in the first antenna of the OSTBC encoder | $t \to j_1(2j-1) = a$ $t + T_s \to j_1(2j) = -b^*$ $j = 1, \dots, 201$ |
| $\vec{j}_{ci,2}$ | Transmission Alamouti coding operation in the second antenna of the OSTBC encoder | $t \to j_2(2j-1) = b$ $t + T_s \to j_2(2j) = a^*$ $j = 1, \dots, 201$ |
| | Modulation | 4-, 8-, *or* 16-*PSK* |
| $SNR$ | Signal-to-Noise Ratio | 0–25 dB |
| $SIR$ | Signal-to-Interference Ratio | −7–25 dB |
| $E_s$ | Energy-to-symbol | 1 (Watt) |
| $S$ | The OSTBC matrix of the transmitter | |

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
