# Peer review of "Enhancing MIMO Spatial-Multiplexing and Parallel-Decoding under Interference by Computational Feedback"

_electronics, doi:10.3390/electronics12030761_

Round 1

Author Response

29/01/2023
Dear Dr. Ms. Ryann Jiang, 
Section Managing Editor At MDPI Electronics!

We are writing you in regards with our paper entitled:
"Utilizing Full Potential MIMO-Spatial Multiplexing Using Computational
Feedback Under General Interference And High-Leveled AWGN"
(Manuscript ID electronics-2185967).
We first wish to thank you for finding and convening qualified reviewers to examine our paper. We would also like to thank you very much for your professional and quick handling of our article.
We thank the reviewers for the encouraging and professional responses that helped us to improve the article.
We have consider all the comments of the reviewers and corrected the article accordingly.
We believe that the readers of the MDPI-Electronics-Microwave and Wireless Communication will be benefited from the elaborated study.

Sincerely,

The authors:

RF Engineer Mr. Avner Elgam,

Dr. for Mathematics and Computer Sciences Yossi Peretz,

Prof. for Communication Systems Yosef Pinhasi.

Reviewer: 1

Comments to electronics-2185967

This paper proposed an alternating projections HSIC algorithm to deal with digital
interference canceling, which enables a parallel decoding process from the parallel
transmission of OSTBC under a complex and challenging wireless environment to
facilitate DSS capability.

Generally, the authors have done a solid work for digital interference cancellation.
Some parts need to be further improved. The reviewer has the following comments:

  • The title of this paper is not precise, as it used “utilizing” and “using”, which has a similar meaning. Replacing “utilizing full” with “enhancing” and “improving” would be better. Besides, “Under General Interference And High-Leveled AWGN” could be removed as it is only the simulation setting.

Thank you for your proposal. We have changed the title accordingly. 

  • Although the authors introduced various methods to address interference-jamming problems, however, the specific introduction of existing works are also needed.

Thank you for your response. We have expanded the sources of information as you requested. We have added seven recent works dealing with jamming attacks and interferences. Four of the works follow your proposal in section 4. Three works are from a repeated review that we did.
Also, in the introduction, we reviewed many of the methods and works done on the subject, including a refinement of the advantages and disadvantages of every technique.

  • The motivation is not well clarified, which is important to demonstrate why the authors investigated this work. The contributions should also be improved to clarify their novelty and potential application.

Thank you for your response. 
In lines 95 to 128, we detailed the advantages of the proposed algorithm, including all the contributions and motivations for developing this critical topic.
Examples of the advantages and contributions of the AP-HSIC that describe in these paragraphs are:
1.  "This computational feedback can overcome the effects of random scatters in a multi-path fading channel and quasi-static Flat-Rayleigh-Fading MIMO channels, combined with high-leveled AWGN and general interference scenarios."
2. "The proposed AP-HSIC can generate significant capabilities for successfully canceling digital interference."
3. "AP-HSIC offers flexibility regarding Stand-Alone (SA) networks without the request to control a sharing channel that statistically measure the spatial domain. The receiver also does not require any control or physical feedback to the transmitter."
4. "AP-HSIC can decode symbols in a parallel MIMO mode in real-time without slowing down the decoding process, simultaneously with the interference cancellation process."

However, following your suggestion, we refined the motivation and the significant contribution of this algorithm with the help of adding a paragraph near the end of the introduction (these lines are highlighted in red).

  • It is suggested to introduce the following recent works in MIMO [R1], interference suppression [R2]-[R3], and NOMA [R4] fields to highlight the state-of-the-art of this paper.

Thank you for your proposal. We expanded and added all the articles you suggested to the article and give references to the article.

  • The MIMO communication model should be drawn and added in section II.

Thank you for your proposal. We have transferred the figure describing the communication model to section II.

  • There are too many symbols in this paper, it is suggested to add a Table to list them.

Thank you for your proposal. Further to your suggestion, we added a table summarizing and describing all the symbols we used in this article.

  • What is the difference between the proposed alternating projections HSIC algorithm and traditional HSIC algorithms? The comparison should be added.

Thank you for your response.
As we wrote in the article, there are two main approaches to dealing with interferences.
The first approach requires physical feedback between the receiver and the transmitter, transmission of wireless channel update information to the transmitter, SVD decomposition or QR decomposition, and construction of orthonormal bases for beam generation and zero steering. With this method, the transmitted information can be decoded simultaneously, but as we mentioned, it requires feedback resources. As we said, the most significant disadvantages of this method are the change in the channel matrix due to disturbances in the decomposition process or transfer of information from the receiver to the transmitter (this process causes inaccuracy in the beam planning), or dispersion of the beams from the scatters/interference in the medium.
The second approach - SIC, does not require feedback. Still, the main assumptions in this method are a low correlation between the desired signal and the other interferers/users or independence between the desired user and the interferers. In addition, it is assumed that the information is transmitted as a transmission block consisting of several sub-blocks to several users or a single user, with the algorithm performing a serial iterative process after building orthonormal bases. As we explained, the main disadvantage of this method is that external interference significantly affects the serial decoding process and causes many decoding errors.
We took a leading technique in the second approach, MGSTC, which produces the most significant diversity gain, as the basis for comparison with the proposed method. 

  • The size of figure 1-3 needs to be enlarged.

Thank you for your response. We immediately corrected Figures 1-3, as you requested.

Reviewer 2 Report

Overall, a well written paper with solid scientific background and mathematical knowledge, which I consider appropiate to be published with small changes of explanatory nature.

Line 4: "The correctness and convergence of the algorithm are proved, and its complexity is given."

I recommend rephrasing for a better understanding: "The correctness and convergence of the algorithm are demonstrated, and an insight of its complexity is provided."

Line 34: "destruction of wireless channels" - please explain what you mean by destruction, maybe a different word would explain it better.

Line 262: "This explains why, in the above-mentioned papers, the simplified cases appear as

the most challenging interferences." - the explanation is unclear to me, maybe some more details in explaining the type of challenges and the impact of the previously written demonstration would help.

Lines 276, 291, 295 etc: achronyms such as S.V.D. or p.d.f. should be written in extenso at least once so that any reader would be able to understand. E.g. The Singular Value Decomposition (S.V.D.) of a matrix etc The same observation is valid also for BER, SNR etc

Line 497: "we present–not only on the MATLAB platform but also on the SIMULINK platform" - maybe "not only on the MATLAB platform but also on the SIMULINK platform" should be put between paranthesis instead of - - 

Figure 2 is extremely small, the blocks are hardly visible with open eye, I had to change the view to 300% to actually see something...

References are actual and reflect authors previous work and research interests in the area of the paper.

Author Response

29/01/2023
Dear Dr. Ms. Ryann Jiang, 
Section Managing Editor At MDPI Electronics!

We are writing you in regards with our paper entitled:
"Utilizing Full Potential MIMO-Spatial Multiplexing Using Computational
Feedback Under General Interference And High-Leveled AWGN"
(Manuscript ID electronics-2185967).
We first wish to thank you for finding and convening qualified reviewers to examine our paper. We would also like to thank you very much for your professional and quick handling of our article.
We thank the reviewers for the encouraging and professional responses that helped us to improve the article.
We have consider all the comments of the reviewers and corrected the article accordingly.
We believe that the readers of the MDPI-Electronics-Microwave and Wireless Communication will be benefited from the elaborated study.

Sincerely,

The authors:

RF Engineer Mr. Avner Elgam,

Dr. for Mathematics and Computer Sciences Yossi Peretz,

Prof. for Communication Systems Yosef Pinhasi.

Reviewer: 2

Overall, a well written paper with solid scientific background and mathematical knowledge, which I consider appropiate to be published with small changes of explanatory nature.

Line 4: "The correctness and convergence of the algorithm are proved, and its complexity is given."
I recommend rephrasing for a better understanding: "The correctness and convergence of the algorithm are demonstrated, and an insight of its complexity is provided."

Thank you for your response.
Regarding the proposed algorithm, we have not only "demonstrated" its proof and gave "insights of its complexity." We actually provide a correctness proof and convergence rate of the proposed algorithm, based on the Alternating-Projections algorithm correctness proof and convergence rate. The use of the general Alternating-Projections algorithm was not able without our formulation of the problem with the connection to the MIMO Communications Simultaneously Interference Cancellation, And Decoding problem, without the definitions of the related closed-convex sets and without the computations of the right projections onto the closed-convex sets. So, the correctness and convergence rate are not self-evident from the general Alternating-Projections algorithm. 
The proof of any new theorem might relay on theorems that were proved before in a previous published references and still the credit for the proof of the new theorem is given to the last, being nontrivial use of the previous results.
However, in order not to take all the credit for ourselves, we have changed the sentence to: "The correctness and convergence of the proposed algorithm are provided, and its complexity is given."

Line 34: "destruction of wireless channels" - please explain what you mean by destruction, maybe a different word would explain it better.

Thank you for your response. We expanded the explanation as you requested. This explanation is detailed in lines 35-38, and highlighted in red.

Line 262: "This explains why, in the above-mentioned papers, the simplified cases appear as the most challenging interferences." - the explanation is unclear to me, maybe some more details in explaining the type of challenges and the impact of the previously written demonstration would help.

Thanks for your response. We expanded and refined the explanation as you requested. This extension appears in lines 278-281 and is highlighted in red.

Lines 276, 291, 295 etc: achronyms such as S.V.D. or p.d.f. should be written in extenso at least once so that any reader would be able to understand. E.g. The Singular Value Decomposition (S.V.D.) of a matrix etc The same observation is valid also for BER, SNR etc

Thank you for your response.  We immediately corrected these issues, as you requested.

Line 497: "we present–not only on the MATLAB platform but also on the SIMULINK platform" - maybe "not only on the MATLAB platform but also on the SIMULINK platform" should be put between paranthesis instead of - - 

Thank you for your response.  We immediately corrected these issues, as you requested.

Figure 2 is extremely small, the blocks are hardly visible with open eye, I had to change the view to 300% to actually see something...

Thank you for your response.  We immediately corrected these issues, as you requested.

References are actual and reflect authors previous work and research interests in the area of the paper.
